# Taming Transformer
# without using learning rate warmup

**Xianbiao Qi**[1], **Yelin He**[1], **Jiaquan Ye**[1], **Chun-Guang Li**[2†], **Bojia Zi**[3], **Xili Dai**[4], **Qin Zou**[5], **Rong Xiao**[1†]
[1]Intellifusion Inc.    [2]BUPT    [3]CUHK    [4]HKUST (GZ)    [5]WHU

## Abstract

Scaling Transformer to a large scale without using some technical tricks such as learning rate warump and using an obviously lower learning rate is an extremely challenging task, and is increasingly gaining more attention. In this paper, we provide a theoretical analysis for the process of training Transformer and reveal the rationale behind the model crash phenomenon in the training process, termed *spectral energy concentration* of $W_q^\top W_k$, which is the reason for a malignant entropy collapse, where $W_q$ and $W_k$ are the projection matrices for the query and the key in Transformer, respectively. To remedy this problem, motivated by *Weyl's Inequality*, we present a novel optimization strategy, *i.e.*, making the weight updating in successive steps smooth—if the ratio $\frac{\sigma_1(\nabla W_t)}{\sigma_1(W_{t-1})}$ is larger than a threshold, we will automatically bound the learning rate to a weighted multiple of $\frac{\sigma_1(W_{t-1})}{\sigma_1(\nabla W_t)}$, where $\nabla W_t$ is the updating quantity in step $t$. Such an optimization strategy can prevent spectral energy concentration to only a few directions, and thus can avoid malignant entropy collapse which will trigger the model crash. We conduct extensive experiments using ViT, Swin-Transformer and GPT, showing that our optimization strategy can effectively and stably train these Transformers without using learning rate warmup.

> *"Nothing in life is to be feared. It is only to be understood."*
> — *Marie Curie*

## 1  Introduction

Transformer (Vaswani et al., 2017) has revolutionized various domains of artificial intelligence, including natural language processing (Radford et al., 2018; 2019; Brown et al., 2020; Chowdhery et al., 2023; Touvron et al., 2023; Dubey et al., 2024) and computer vision (Dosovitskiy et al., 2020; Liu et al., 2021) and many more applications (Radford et al., 2021; Ramesh et al., 2021; Peebles & Xie, 2023), owning to their ability to capture long-range dependencies through self-attention mechanisms. However, despite their widespread application and empirical success, training deep Transformer models remains quite challenging. Practitioners frequently encounter variant issues, such as gradient explosion (Qi et al., 2023b), rank collapse (Dong et al., 2021), entropy collapse (Zhai et al., 2023) and general training instability (Kim et al., 2021; Qi et al., 2023b), especially during the initial training stage.

To address these challenges, researchers have proposed various modifications to the original Transformer architecture, including altering the placement of Layer Normalization (Wang et al., 2019; Xiong et al., 2020) (*e.g.*, pre-LN vs. post-LN schemes), carefully conditioning the residual connections (Bachlechner et al., 2021), and QKNorm (Henry et al., 2020; Dehghani et al., 2023) for the self-attention module. Similarly, DeepNet (Wang et al., 2022) introduces a new normalization function to modify the residual connection in Transformer. ReZero (Bachlechner et al., 2021) introduces a learnable residual scalar parameter for the residual shortcut, and requires initiating it to 0 at the start stage of training. More recent approaches (Kim et al., 2021; Qi et al., 2023a) have focused on examining and enforcing Lipschitz continuity properties of Transformer

---

† Corresponding authors

components, which can provide insights into the network behavior and the training stability. *Although there are a few works (Bachlechner et al., 2021; Qi et al., 2023a) that can avoid using learning rate warmup to train Transformer successfully, all of them require significant modifications of the network structure.*

*Learning rate warmup* (Loshchilov & Hutter, 2016) seems to be a must-have technology for standard optimizers (Robbins & Monro, 1951; Duchi et al., 2011; Kingma & Ba, 2014; Loshchilov & Hutter, 2019) in some popular large Transformer models (Radford et al., 2018; 2019; Brown et al., 2020; Chowdhery et al., 2023; Touvron et al., 2023). Without the learning rate warmup stage, the Transformer training will prone to diverge.

Although it is usual to train a Transformer by modifying the network structure as mentioned above or using the learning rate warmup, two natural and interesting questions remain:

1. *What are the training dynamics of a Transformer model when its training fails or succeeds?*
2. *Can we successfully tame a Transformer without changing the network structure or without using learning rate warmup?*

This paper aims to answer these questions. To answer the first question, we examine the training processes of three types of Transformers, by visualizing the changing trajectories along with the training process of 15 (or 13) quantities about the parameters, activations, and attention maps. By doing so, we observe that the model crash is accompanied by a weird phenomenon that the entropy of the attention map is almost 0 and the spectral norm of $W_q{}^\top W_k$ increases to a very large value. By conducting mathematical analysis for the Transformer training, we identify that the Spectral Energy Concentration (SEC) of $W_q{}^\top W_k$ is the key problem leading to the model crash. To answer the second question, motivated by Weyl' Inequality, we present a novel optimization strategy, *i.e.*, making weight updating smooth, and verify empirically that our optimization strategy can prevent spectral energy concentration and thus achieving a stable convergence in training.

**Paper Contributions.** The contributions of the paper are highlighted as follows.

- We visualize the training dynamics of Transformers that train successfully or unsuccessfully and summarize two important observations from unsuccessful training that: a) the rank of the attention map matrix tends to very low and the entropy of attention probability matrix tends to 0; and b) $\sigma_1(W_q{}^\top W_k)$ increases rapidly to a very large value.
- We present theoretical analysis for the Transformer training, finding that the Jacobian matrix $\frac{\partial \text{vec}(P)}{\partial \text{vec}(W_q{}^\top W_k)} = X^\top \otimes X^\top$, where $P = X^\top W_q{}^\top W_k X$. It implies that the gradient of $W_q{}^\top W_k$ is largely dominated by the rank of $X^\top \otimes X^\top$.
- We reveal that SEC of $W_q^\top W_k$ makes the attention map matrix to be sparse yet low-rank and it is the reason leading to model crash.
- Motivated by the Weyl's inequality, we introduce a novel strategy to address the problem of SEC of $W_q{}^\top W_k$ by controlling the rapid growth of singular values, and verify that our strategy leads to a stable training process.

## 2 PRELIMINARIES

**Matrix norm.** Given a matrix $W$, its $\ell_p$-norm is defined as: $\|W\|_p = \sup_{x \neq 0} \frac{\|Wx\|_p}{\|x\|_p}$. When $p = 2$, the induced matrix norm is the *spectral norm*, which is defined as the largest singular value of $W$ and is also expressed as the square root of the largest eigenvalue of the Gram matrix $W^\top W$. The spectral norm of a matrix $W$ can be calculated as: $\|W\|_2 = \max_{x \in S^{n-1}} \|Ax\|_2 = \sqrt{\lambda_{\max}(W^\top W)} = \sigma_1(W)$, where $\sigma_1(W)$ denotes the largest singular value of matrix $W$ and $\lambda_{\max}(W^\top W)$ denotes the largest eigenvalue of $W^\top W$, and $S^{n-1}$ denotes a unit sphere in $\mathcal{R}^n$.

**Power iteration to compute matrix spectral norm.** The power iteration algorithm starts with a vector $x_0$ of unit $\ell_2$-norm. The entire iteration process is as follows: $x_{k+1} = \frac{Wx_k}{\|Wx_k\|_2}$ for $k = 0, \cdots, K-1$. At every iteration, $x_k$ is multiplied by matrix $W$ and normalized. After $K$ iterations,

$\|\boldsymbol{x}_K\|_2$ is used as the estimated spectral norm. Usually, it takes 3 to 5 iterations to converge, and thus the computation cost is cheap.

**Adam Optimizer.** Adam (Kingma & Ba, 2014) is currently the most widely used optimizer for training neural networks, owing to its efficiency and effectiveness. Adam can be simply defined as: $\boldsymbol{M}_t = \beta_1 \boldsymbol{M}_{t-1} + (1-\beta_1)\boldsymbol{G}_t$, $\quad \boldsymbol{V}_t = \beta_2 \boldsymbol{V}_{t-1} + (1-\beta_2)\boldsymbol{G}_t^2$, $\quad \boldsymbol{W}_t = \boldsymbol{W}_{t-1} - \alpha_t \boldsymbol{M}_t \oslash \sqrt{\boldsymbol{V}_t + \epsilon}$, where $\boldsymbol{G}_t$ is the gradient at step $t$, $\boldsymbol{G}_t^2$ is the element-wise square of $\boldsymbol{G}_t$, and $\oslash$ denotes element-wise division, $\alpha_t$ is the learning rate at step $t$, and $\beta_1, \beta_2$ are the first-order and the second-order momentum factors, respectively.

## 3 TAMING TRANSFORMER REQUIRES TO REVISIT THE TRAINING DYNAMICS

To begin with, we first give some basic notions in a Transformer, which includes an attention module, an FFN module and two normalization modules that are used before the attention module and the FFN module. For the attention module, we usually use a multi-head attention which allows the model to jointly attend to information from different representations with different heads. Here, for the convenience, without losing generality, we only use a single-head attention. To be precise, we define each of them as follows:

$$
\begin{aligned}
\text{Attn}(\boldsymbol{X}; \boldsymbol{W}_q, \boldsymbol{W}_k, \boldsymbol{W}_v, \boldsymbol{W}_o) &= \boldsymbol{W}_o \boldsymbol{W}_v \boldsymbol{X} \,\text{softmax}\left( \frac{\boldsymbol{X}^\top \boldsymbol{W}_q^\top \boldsymbol{W}_k \boldsymbol{X}}{\sqrt{d_q}} \right), \\
\text{FFN}(\boldsymbol{x}; \boldsymbol{W}_1, \boldsymbol{W}_2) &= \boldsymbol{W}_2 \,\text{ReLU}(\boldsymbol{W}_1 \boldsymbol{x}), \\
\text{LN}(\boldsymbol{x}) &= \boldsymbol{\gamma} \odot \boldsymbol{z} + \boldsymbol{\beta}, \quad \text{where } \boldsymbol{z} = \frac{\boldsymbol{y}}{\text{std}(\boldsymbol{y})} \text{ and } \boldsymbol{y} = \left( \boldsymbol{I} - \frac{1}{D}\mathbf{1}\mathbf{1}^\top \right)\boldsymbol{x},
\end{aligned}
$$

where $\boldsymbol{X} \in \mathcal{R}^{d \times n}$, $\boldsymbol{W}_q \in \mathcal{R}^{d_q \times d}$, $\boldsymbol{W}_k \in \mathcal{R}^{d_q \times d}$, $\boldsymbol{W}_v \in \mathcal{R}^{d_v \times d}$, $\boldsymbol{W}_o \in \mathcal{R}^{d \times d}$, $\boldsymbol{W}_1 \in \mathcal{R}^{4d \times d}$, $\boldsymbol{W}_2 \in \mathcal{R}^{d \times 4d}$, $\boldsymbol{\gamma} \in \mathcal{R}^d$ and $\boldsymbol{\beta} \in \mathcal{R}^d$. Note that only in a single-head definition, $\boldsymbol{W}_o$ can be put before $\boldsymbol{W}_v$; otherwise, it should be after a concatenation operator. For the convenience of discussion, we define:

$$
\boldsymbol{A} = \text{softmax}(\frac{\boldsymbol{P}}{\sqrt{d_q}}) \quad \text{and} \quad \boldsymbol{P} = \boldsymbol{X}^\top \boldsymbol{W}_q^\top \boldsymbol{W}_k \boldsymbol{X},
$$

where $\boldsymbol{A}$ is called the attention map and $\frac{\boldsymbol{P}}{\sqrt{d_q}}$ is usually called the logit.

### 3.1 VISUALIZATION: WHAT HAPPENS WHEN A TRANSFORMER TRAINING FAILS OR SUCCEEDS

Visualizations are commonly used as an effective means to help us examine whether the neural network's training succeeds or fails. In particular, what happens when the training of a Transformer tends to crash? And what happens when the training succeeds?

One of the most important aspects of understanding the training of neural networks is the observation of changes in parameters and activations. Since the parameters or activations and their gradients are matrices or vectors, the norm is the best way to observe these quantities. In this paper, for a Transformer training, we summarize the following **15 terms** to watch:

$$
\begin{aligned}
& \sigma_1(\boldsymbol{W}_q), \quad \sigma_1(\boldsymbol{W}_k), \quad \sigma_1(\boldsymbol{W}_v), \quad \sigma_1(\boldsymbol{W}_o), \quad \sigma_1(\boldsymbol{W}_1), \quad \sigma_1(\boldsymbol{W}_2), \quad \sigma_1(\boldsymbol{W}_q^\top \boldsymbol{W}_k), \\
& \sigma_1(\boldsymbol{W}_o \boldsymbol{W}_v), \quad \|\boldsymbol{\gamma}_1\|_2, \quad \|\boldsymbol{\beta}_1\|_2, \quad \|\boldsymbol{\gamma}_2\|_2, \quad \|\boldsymbol{\beta}_2\|_2, \quad \sigma_1(\boldsymbol{W}_2 \boldsymbol{W}_1), \quad \|\boldsymbol{x}\|_2, \quad \left\|\frac{\partial L}{\partial \boldsymbol{x}}\right\|_2,
\end{aligned}
\tag{1}
$$

where $\boldsymbol{\beta}_1$ and $\boldsymbol{\gamma}_1$ denote the parameters of the first LayerNorm, and $\boldsymbol{\beta}_2$ and $\boldsymbol{\gamma}_2$ denote the parameters of the second LayerNorm. When RMSNorm (Zhang & Sennrich, 2019) is used, there are only 13 terms that will be analyzed since it does not have $\boldsymbol{\beta}$. For the weight matrix, we use the spectral norm. For a vector, we use its $\ell_2$ norm.

To ensure that the phenomena we observed can generalize well, we visualized them on both ViT (Dosovitskiy et al., 2020) and GPT (Radford et al., 2018). ViT is a pure encoder architectures whereas GPT is a pure decoder architecture.

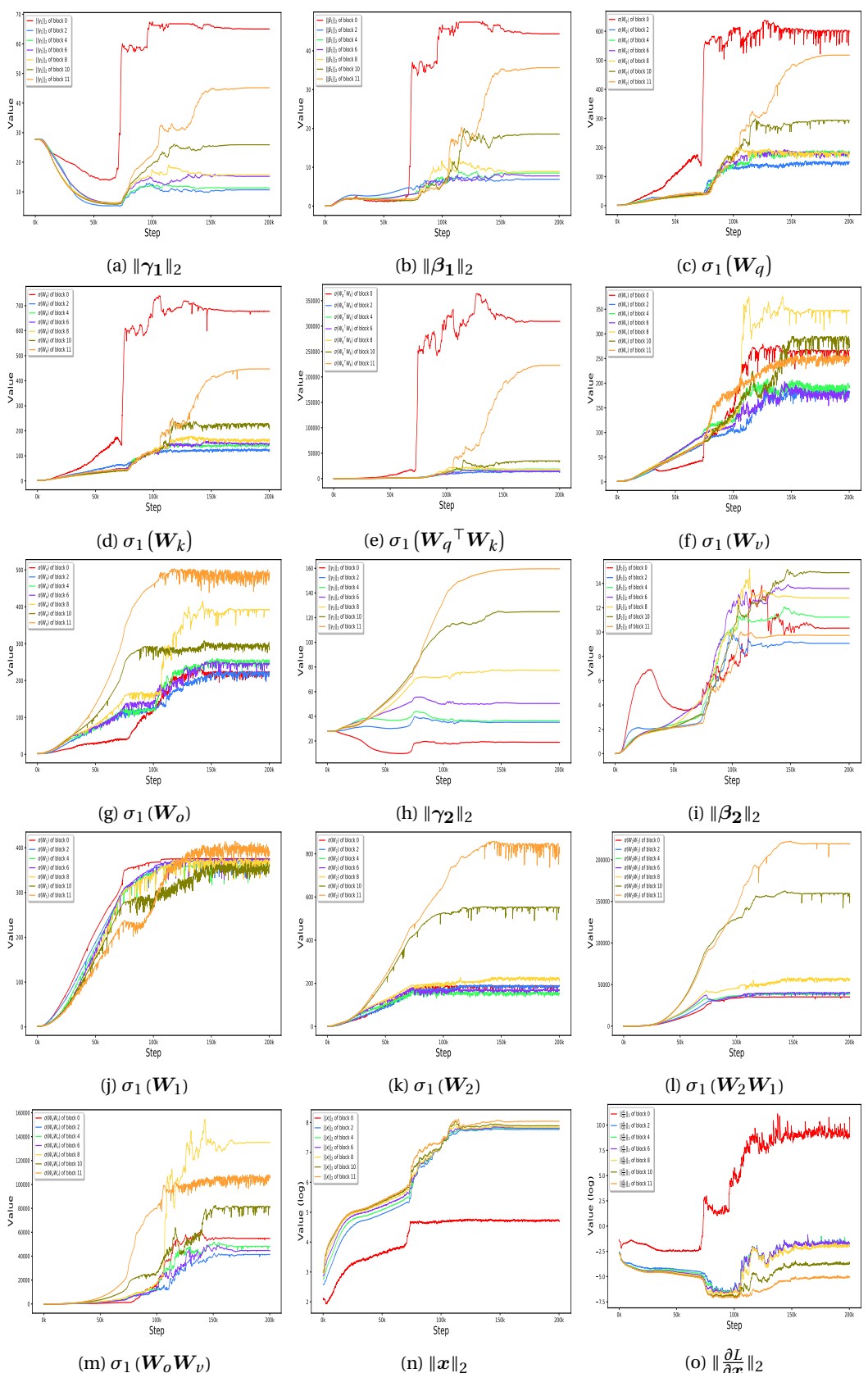

FIGURE 1: Training dynamics of a failure ViT. This figure shows how 15 items of quantities as defined in Equation (1) change during the training period. Please pay more attention to subfigures (a)-(e).

(a) *Block 0 (successful).*    (b) *Block 6 (successful).*    (c) *Block 11 (successful).*

(d) *Block 0 (crashed).*    (e) *Block 6 in (crashed).*    (f) *Block 11 (crashed).*

FIGURE 2: Visualization of the dynamics process of attention map in different training steps for a successful and a crashed ViT-base model, respectively. *Please click the images to play the flash. Best viewed with Acrobat Reader.*

In this section, due to the space limitation, we will only visualize ViT-base, and put more visualization results into the Appendix H. For the ViT implementation, we use Timm Wightman (2019), in which "timm" library provides rich model architectures of many pre-trained image models in PyTorch. For the GPT implementation, we use nanoGPT, which uses LayerNorm without a bias term, and thus only watch 13 terms, rather than 15 terms in ViT.

To achieve a successful ViT training, we use a long learning rate warmup. For instance, we use 60 epochs of warmup and the whole training process takes 150 epochs. To obtain the dynamics of a failure training of ViT, we do not use warmup.

Figure 1 visualizes a failure training process of a ViT-base model. The model includes 12 blocks with index from 0 to 11. In Figure 1, we visualize the weight matrices in blocks $\{0, 2, 4, 6, 8, 10, 11\}$, where the index of the last block is 11. For the features $x$ and the gradients $\frac{\partial L}{\partial x}$, we hook the input features that enter into the corresponding blocks. Figure 8 in the Appendix I visualizes a successful training process of a ViT-L model. Meanwhile, in Figure 2, we visualize the dynamic processes of attention maps during the training period about a successful case and a failure case of the ViT-base model, respectively. We visualize the attention map of the same image at different steps.

We observe the following phenomena from Figures 1 and 2.

- As shown in Figure 1, at the beginning, the maximum singular value $\sigma_1(W_q^\top W_k)$ gradually increases, and at a certain point, the maximum singular value suddenly and rapidly increases to a very large value (*e.g.*, around 200,000), at where the loss divergence emerges. However, for a successful training process, $\sigma_1(W_q^\top W_k)$ gradually increases to a medium value as shown in Figure 8 and then vibrates around that value.

- As shown in Figure 2, in a failure training process of Transformer, the attention maps gradually become sparse and low-rank. Finally, the entropy of the attention map is 0; whereas in a successful training process, the attention map is not too sparse and of a medium rank.

- The normalization layers exhibit a huge difference: $\gamma_1$ and $\beta_1$ in a successful ViT training process are very smooth, but $\|\gamma_1\|_2$ and $\|\beta_1\|_2$ suddenly increase dramatically in an unsuccessful case.

- The ranges of the activation and the gradients are very large in a crashed model, and the gradients in different blocks vary much larger than that in a successful model.

**Remark.** We summarize that the successful training and the unsuccessful training of a Transformer exhibit significant differences among their $\sigma_1(W_q^\top W_k)$, their normalization parameters $\gamma$ and $\beta$, and their activations $x$ and gradients $\frac{\partial L}{\partial x}$.

## 3.2 THEORETICAL ANALYSIS: MATRIX CALCULUS OF TRANSFORMER

To understand the training dynamics of Transformer, we should investigate the process of back-propagation (Rumelhart et al., 1986; LeCun et al., 2002; 1989; 1998). In the attention mechanism, however, the input and the output are both matrices, we cannot directly use vector calculus. Instead, we need to use *Vectorization* (Graham, 2018; Petersen et al., 2008) and *Kronecker Product* (Graham, 2018; Petersen et al., 2008). In matrix calculus, the vectorization of a matrix is a linear transformation that converts a matrix into a vector. Specifically, the vectorization of a matrix $M \in \mathcal{R}^{m \times n}$, denoted as $\text{vec}(M)$, is a column vector, obtained by an ordered stacking of the columns of the matrix $M$, i.e., $\text{vec}(M) \in \mathcal{R}^{mn}$. For example, for a $2 \times 3$ matrix $M = \begin{bmatrix} a & b & c \\ d & e & f \end{bmatrix}$, the vectorization of $M$ is $\text{vec}(M) = [a\ d\ b\ e\ c\ f]^\top$. For the attention module of Transformer, we have the following proposition about the Jacobian matrix of the output $P$ with respect to the input $X$ and the parameters.

**Proposition 1 (Matrix Calculus for Self-Attention)**
*Let $P = X^\top W_q^\top W_k X$, where $X \in \mathcal{R}^{d \times n}, W_q \in \mathcal{R}^{d_q \times d}, W_k \in \mathcal{R}^{d_q \times d}$, according to vectorization and matrix calculus, we have the following derivations:*

$$\frac{\partial \text{vec}(P)}{\partial \text{vec}(W_q^\top W_k)} = X^\top \otimes X^\top, \quad \frac{\partial \text{vec}(P)}{\partial \text{vec}(X)} = (X^\top W_k^\top W_q \otimes I_n)K + (I_n \otimes X^\top W_q^\top W_k), \quad (2)$$

$$\frac{\partial \text{vec}(P)}{\partial \text{vec}(W_q^\top)} = (W_k X)^\top \otimes X^\top, \quad \frac{\partial \text{vec}(P)}{\partial \text{vec}(W_k)} = X^\top \otimes (W_q X)^\top, \quad (3)$$

*where $\otimes$ denotes the Kronecker product, $I_n \in \mathcal{R}^{n \times n}$ denotes an identity matrix with shape $n \times n$, $K$ is the commutation matrix, which depends on the dimensions of $X$. Since $X \in \mathcal{R}^{d \times n}$, then we know $K \in \mathcal{R}^{nd \times nd}$. The commutation matrix $K$ has the property that $\text{vec}(X^\top) = K \text{vec}(X)$ for any matrix $X$.*

In Appendices A and B, we supply some elementary information for the vectorization and the Kronecker product and the derivations of the Jacobian matrix for a single-head attention.

We have the following observations from Proposition 1.

- We have $\frac{\partial \text{vec}(P)}{\partial \text{vec}(W_q^\top W_k)} = X^\top \otimes X^\top$ in Equation 2, and we know about the Kronecker product that $\text{rank}(X^\top \otimes X^\top) = \text{rank}(X^\top)^2$, which implies that if the rank of $X$ is very low, then the rank of $\frac{\partial \text{vec}(P)}{\partial \text{vec}(W_q^\top W_k)}$ will also be very low. Note that $X$ being low rank means that the features across different timestep are highly correlated or coherent. If all $x_i$ in $X$ collapses to a single point, then $X^\top \otimes X^\top$ will only have a large singular value, and the rest are 0.

- The Jacobian matrix $\frac{\partial \text{vec}(P)}{\partial \text{vec}(X)}$ in Equation 2, is in direct proportion to $X$ and $W_q^\top W_k$. If the spectral norm $\sigma_1(W_q^\top W_k)$ is very large, it implies that the gradient $\frac{\partial L}{\partial X}$ will more likely to be magnified a lot.

- Equation 3 suggests that changes in the query weights $W_q$ are related to both the input $X$ and the key representation $W_k X$. Equation 3 suggests that changes in the key weights $W_k$ are related to both the input $X$ and the query representation $W_q X$.

- All these relationships are interconnected, with changes in one variable potentially affecting the others. For instance, if $W_k$ increases fast, then according to Equation 3, $\frac{\partial \text{vec}(P)}{\partial \text{vec}(W_q^\top)}$ will more likely to be very large. In this way, $W_q$ will likely increase very fast.

## 3.3 RATIONALE IN MODEL CRASH: SPECTRAL ENERGY CONCENTRATION

Before we reveal the rationale in model crash, let us first discuss a bit on the entropy collapse.

**Two Entropy Collapse Modes.** In experiments, we observe two types of attention entropy collapse modes. Note that when attention collapse happens, the attention map tends to

---

```
https://en.wikipedia.org/wiki/Vectorization_(mathematics)
https://en.wikipedia.org/wiki/Kronecker_product
```

a sparse matrix (*i.e.*, there are a few dominate nonzero coefficients in attention map), and thus the entropy of the attention map is vanishing. To be more specific, when the attention map is sparse but not low-rank, we call it a *benign collapse*; whereas if the attention map is sparse and simultaneously low-rank, we call it a *malignant collapse*. When benign collapse occurs, the attention map is shown in the right panel of Figure 3 that, there is almost an identity matrix. In this way, the diagonal elements are almost 1, and the non-diagonal elements have values around 0. Unfortunately, when malignant collapse happens, the attention map becomes a sparse and simultaneously low-rank matrix as shown in the middle panel of Figure 3. Furthermore, we observe that the distribution of the spectral energy of $W_q{}^\top W_k$ for the benign collapse is relatively uniform; whereas the spectral energy of the attention matrix for the malignant collapse tends to concentrate on a few dominate singular values.





*(a) Normal attention*   *(b) Malignant collapse*   *(c) Benign collapse*



FIGURE 3: Three modes of attention maps. The left panel shows a normal attention map. The middle panel shows a classical attention map when model crash occurs, for which the entropy is almost 0. The right panel shows an attention map from a normal model training but its entropy is almost 0.

By analyzing the matrix $W_q{}^\top W_k$ in the benign collapse when it happens in the experiments, we find that it has the following property: $W_q{}^\top W_k$ is usually a non-symmetric positive quasi-definite square matrix (see Appendix O for details), and the self-attention layer degenerates into a linear projection layer because

$Y = W_v X A \approx W_v X I = W_v X$. We give an intuitive analysis in Appendix C.

When the malignant collapse happens, the model will usually crash. We identify that the rationale behind the model crash is a phenomenon called *spectral energy concentration* (*SEC*). Before we present our theorem about SEC, let us first introduce an index to quantify it. Recall that $W_q \in \mathcal{R}^{d_q \times d}$ and $W_k \in \mathcal{R}^{d_q \times d}$, where $d_q < d$. We have that $W_q{}^\top W_k \in \mathcal{R}^{d \times d}$, but its rank is less than or equal to $d_q$. Precisely, we define a *SEC index* as follows:

$$\text{SEC}(d_q, s) = \frac{\sum_{i=1}^{s} \sigma_i^2(W_q{}^\top W_k)}{\sum_{i=1}^{d_q} \sigma_i^2(W_q{}^\top W_k)}, \tag{4}$$

where $d_q$ is the head dimension and $s \le d_q$. For instance, if we have $d_q = 64$ and $s = 4$, and if at this time, $\text{SEC}(64, 4) > 99\%$, we could say the spectral energy of $W_q{}^\top W_k$ highly concentrates on only four dominant singular values.

To be precise, we have the following theorem for the reason to cause a malignant collapse.

**Theorem 1 (Malignant Entropy Collapse)**
*Let* $P = X^T W X$ *and* $A = \text{softmax}\left(\dfrac{P}{\sqrt{d_q}}\right)$, $W = W_q^T W_k \in \mathbb{R}^{d \times d}$. *Suppose that the following two conditions are simultaneously satisfied:*

- $X$ *is a low-rank matrix;*

- $W$ *is a low-rank matrix with only a few dominant singular values (e.g., the singular values are greater than* $C_0 \cdot \sqrt{d_q}$ *where* $C_0 \gg 1$ *is a constant).*

*Then, the attention map* $A$ *will be sparse and simultaneously low-rank in high probability.*

When the malignant entropy collapse happens, the training process will crash. We provide a proof for Theorem 1 in Appendix D.

---

For a sparse and simultaneously low-rank matrix, we refer the reader to a recent textbook (Wright & Ma, 2022).

According to Theorem 1, we know that the model crash is caused by the spectral energy concentration. In Figure 4, we evaluate and compare the curves of SEC($d_q$, $s$) of three different blocks under a successfully trained model and a crashed model. In a successfully trained model, the spectral energy distributes in all directions. However, the spectral energy of a crashed model only concentrates on a few directions. As shown in Figure 4, in a crashed model, the SEC collapses into less than 10 directions. Figure 5 reveals how the attention collapse propagates through

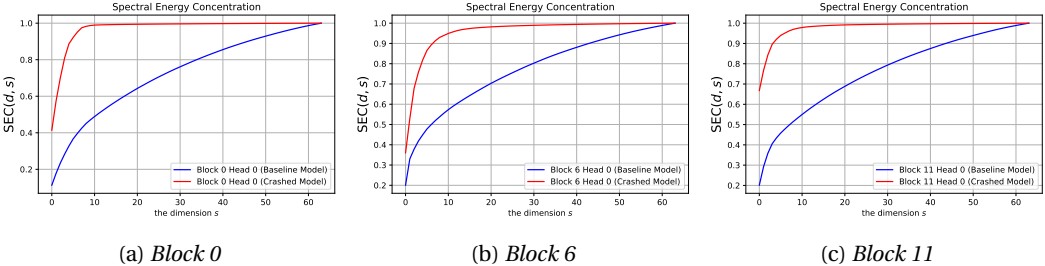

(a) *Block 0*        (b) *Block 6*        (c) *Block 11*

FIGURE 4: Comparison of spectral energy concentration index SEC between a successfully trained model and a crashed model. The attention maps of three different blocks are shown. The spectral energy distributes in all directions in a successful training case; whereas the spectral energy only concentrates on a few directions in a crashed model.

$$X^l{\downarrow} \longrightarrow X^\top \otimes X^\top{\downarrow} \longrightarrow \frac{\partial \text{vec}(P)}{\partial \text{vec}(W_q^\top W_k)}{\downarrow} \longrightarrow W_q^\top W_k{\downarrow} \longrightarrow A{\Downarrow} \longrightarrow X^{l+1}{\downarrow}$$

FIGURE 5: Attribution flow chart of attention collapse.

each term, illustrating the entire attribution chain from the input $X^l$ to the output $X^{l+1}$ in the attention module. Note that $\downarrow$ indicates being low-rank and $\Downarrow$ means being sparse and simultaneously low-rank. If $X$ is low-rank, then $X^\top \otimes X^\top$ is also low-rank because rank($X \otimes X$) = rank($X$) · rank($X$). According to the gradient computation, we have $\frac{\partial \text{vec}(P)}{\partial \text{vec}(W_q^\top W_k)} = X^\top \otimes X^\top$, thus we know $\frac{\partial \text{vec}(P)}{\partial \text{vec}(W_q^\top W_k)}$ is also low-rank. Meanwhile, it should be noted that the spectral energy of $\frac{\partial \text{vec}(P)}{\partial \text{vec}(W_q^\top W_k)}$ is over-concentrated, which means that the gradient update will dramatically change $W_q^\top W_k$, thus $W_q^\top W_k$ will have large probability to be low-rank. In the paper, we have proved that being low-rank and the leading singular values of $W_q^\top W_k$ are very large will lead to the attention map $A$ over-concentrated (see Appendix C for the proof), and become to a sparse and simultaneously low-rank matrix. Finally, an over-concentrated $A$ will lead to $X^{l+1}$ to be low-rank. We provide a more detailed analysis of Figure 5 in Appendix Q.

### 3.4 OUR SOLUTION: TAMING TRANSFORMER VIA WEYL'S INEQUALITY

The analysis above reveals that spectral energy concentration is the key reason leading to unstable training. One manifestation of spectral energy concentration is the rapid growth of the singular values of of the weight matrices. Thus, our basic idea to prevent malignant collapse is to suppress the fast growth of the singular values. Fortunately, Weyl's inequality provides us a simple but effective tool.

**Theorem 2 (Weyl's Inequality on Singular Values)**
*Let $W_1, W_2 \in \mathcal{R}^{m \times n}$ where $m \geq n$, $\sigma_1(W_1) \geq \sigma_2(W_1) \geq ... \geq \sigma_n(W_1)$ be the ordered singular values of $W_1$, and $\sigma_1(W_2 \geq \sigma_2(W_2) \geq ... \geq \sigma_n(W_2)$ be the ordered singular values of $W_2$. Then we have:*

$$\sigma_{i+j-1}(W_1 + W_2) \leq \sigma_i(W_1) + \sigma_j(W_2).$$

The proof for Theorem 2 is provided in Appendix E. From Theorem 2, it is easy to see that $\sigma_1(W_1 + W_2) \leq \sigma_1(W_1) + \sigma_1(W_2)$. Let $W_t$ be the weight matrix at time step $t$, $\nabla W_t$ be the quantity computed from the gradients and their derivations (where $\nabla W_t$ can be obtained by

---

**Algorithm 1** AdamW$^2$: Taming Transformer via Weyl' Inequality without learning rate warmup.

---

Input: learning rate scheduler $\alpha_t$, weight decay $\lambda$, and first-order and second-order momentum $\beta_1$, $\beta_2$
Output: updated weight $w_T$

1: **for** $t = 1, 2, \ldots, T$ **do**
2:     $G_t = \nabla_{W_{t-1}} \mathcal{L}$
3:     $M_t = \beta_1 M_{t-1} + (1 - \beta_1) G_t, \quad V_t = \beta_2 V_{t-1} + (1 - \beta_2) G_t^2$
4:     $\hat{M}_t = M_t / (1 - \beta_1^t), \quad \hat{V}_t = V_t / (1 - \beta_2^t)$
5:     $\nabla W_t = \hat{M}_t \oslash \sqrt{\hat{V}_t + \epsilon}$                                      ▷ $\nabla W_t$ is the final update quantity.
6:     $\hat{\delta}_1 = \text{PowerIter}(\nabla W_t), \quad \hat{\sigma}_1 = \text{PowerIter}(W_{t-1})$      ▷ Power iteration to compute spectral norm.
7:     **if** $\frac{\alpha_t \hat{\delta}_1}{\hat{\sigma}_1} > \tau$ **then**                     ▷ If Rule 1 does not meet, adjust the learning rate $\alpha_t$.
8:         $\alpha_t = \frac{\tau \hat{\sigma}_1}{\hat{\delta}_1}$
9:     **end if**
10:    **if** Weight Decay is Yes **then**
11:       $W_t = W_{t-1} - \alpha_t \nabla W_t - \alpha_t \lambda_t W_{t-1}$
12:    **else**
13:       $W_t = W_{t-1} - \alpha_t \nabla W_t$
14:    **end if**
15: **end for**

---

SGD (Robbins & Monro, 1951), Adagrad (Duchi et al., 2011), or Adam (Kingma & Ba, 2014)), $\alpha_t$ is the learning rate at time step $t$. Usually, our update equation is $W_t = W_{t-1} - \alpha_t \nabla W_t$. According to Weyl's Inequality, we have,

$$\sigma_1(W_t) = \sigma_1(W_{t-1} - \alpha_t \nabla W_t) \leq \sigma_1(W_{t-1}) + \alpha_t \sigma_1(\nabla W_t). \tag{5}$$

An important observation from Equation 5 is that if $\sigma_1(\nabla W)$ is very large, then $W_t$ will be significantly different from $W_{t-1}$. It means that the successive updates of $W_t$ at time step $t$ from time step $t-1$ would "*jump*" too much. A "*smoother*" updating should satisfy the following rule.

**Rule 1 (Steady Weights Updating Rule)**
*Given weights matrix $W_{t-1}$ and the updating quantity $\nabla W_t$ at step $t$, with the learning rate $\alpha_t$, a steady weights should satisfy the following inequality: $\|W_{t-1} - \alpha_t \nabla W_t\|_2 \leq (1 + \tau)\|W_{t-1}\|_2$, where $\tau > 0$ is a small factor.*

To meet Rule 1, what we need is that $\sigma_1(W_{t-1}) + \alpha_t \sigma_1(\nabla W_t) \leq (1 + \tau)\|W_{t-1}\|_2 = (1 + \tau)\sigma_1(W_{t-1})$. It is easy to see that:

$$\alpha_t \leq \tau \frac{\sigma_1(W_{t-1})}{\sigma_1(\nabla W_t)}. \tag{6}$$

This inequality tell us that the learning rate $\alpha_t$ should be bounded by a ratio of singular values $\sigma_1(W_{t-1})$ and $\sigma_1(\nabla W_t)$. Generally, $\tau$ is a small value, *e.g.*, 0.004 or 0.005. The intuition behind is that if the spectral norm of $\nabla W_t$ is significantly larger than that of $W_t$, then the model is potentially undergoing rapid changes. In such cases, a large learning rate could lead to training instability. Therefore, our strategy is that if $\alpha_t > \tau \frac{\sigma_1(W_{t-1})}{\sigma_1(\nabla W_t)}$ then we truncate $\alpha_t$ to $\tau \frac{\sigma_1(W_{t-1})}{\sigma_1(\nabla W_t)}$. Since our base optimizer is AdamW (Loshchilov & Hutter, 2019) and our algorithm is motivated by Weyl's Inequality, we term our algorithm as AdamW$^2$.

For clarity, we summarize our AdamW$^2$ in Algorithm 1, where lines 6-9 highlight our improvements over the base optimizer, the other codes are same as AdamW. According to line 6 in Algorithm 1, $\sigma_1(\nabla W_t)$ and $\sigma_1(W_{t-1})$ are computed via a fast power iteration method. In practice, we set the maximum iterations in power iteration to 3. Actually, we find that two iterations are enough to estimate the spectral norm of matrices. According to Equation 6, if $\alpha_t > \tau \frac{\sigma_1(W_{t-1})}{\sigma_1(\nabla W_t)}$, then the learning rate $\alpha_t$ will be truncated to $\tau \frac{\sigma_1(W_{t-1})}{\sigma_1(\nabla W_t)}$, or the algorithm will adjust $\alpha_t$ and use the default learning rate set by the learning rate schedule.

## 4 EXPERIMENTS

Compared to some previous works (Bachlechner et al., 2021; Wang et al., 2019; Xiong et al., 2020; Wang et al., 2022; Qi et al., 2023b) that focus on improving the training stability of Transformer,

TABLE 1: Quantitative comparison of AdamW and AdamW$^2$ with and without learning rate warmup. AdamW$^2$ demonstrates a very competitive performance compared to AdamW.

| Method | ViT (Acc. ↑) | | GPT (Loss ↓) | Swin-Transformer (Acc. ↑) | |
|---|---|---|---|---|---|
| Configurations | ViT-B | ViT-L | GPT-S | Swin-S | Swin-B |
| Parameters | 86M | 307M | 125M | 50M | 88M |
| AdamW (with warmup) | 80.22 | 81.65 | 2.848 | 83.02 | 83.48 |
| AdamW$^2$ (no warmup) | 80.58 | 81.82 | 2.840 | 83.14 | 83.44 |

our AdamW$^2$ does not need to adjust the network structure and we do not use learning rate warmup. For ViT, Swin-Transformer and GPT, we will use a warmup of 60 epochs, 20 epochs, and 2000 steps, respectively. In AdamW$^2$, we directly use a cosine learning rate schedule and decay the learning rate from maximum to minimum.

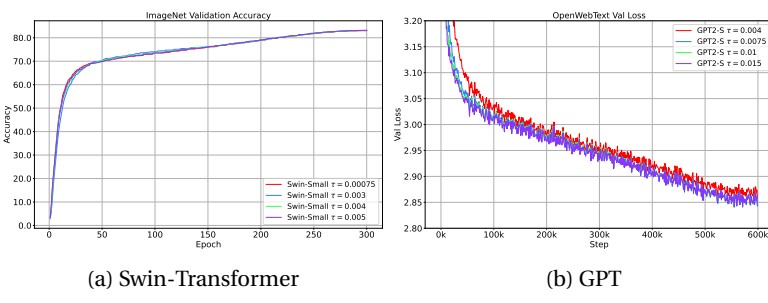

(a) Swin-Transformer        (b) GPT

FIGURE 6: Ablation study of $\tau$ in AdamW$^2$ using Swin-S and GPT-S.

We conduct experiments on three popular Transformers, *i.e.*, ViT (Dosovitskiy et al., 2020), GPT-2 (Radford et al., 2019) and Swin-Transformer (Liu et al., 2021), where ViT and Swin-Transformer are pure encoder architectures and GPT is a pure causal decoder. Note that we do not conduct any adjustments to the networks and directly use the original implementation. Our experiments include image classification on ImageNet (Deng et al., 2009) and large language model on OpenWebText (Gokaslan & Cohen) dataset. We list some training configurations in Appendix N. The quantitative results are shown in Table 1. Our baseline model is the corresponding Transformer using a learning rate warmup; whereas baseline models without using learning rate warmup will crash. AdamW$^2$ demonstrates a very competitive performance compared to the baseline method. These experimental results verify that our understanding of the training dynamics of the Transformer is rational.

We also conduct an ablation study of the choice of $\tau$ in GPT and Swin-Transformer. The results are shown in Figure 6. We can see that the performance of AdamW$^2$ varies slightly for different values of $\tau$, but overall, our approach is robust for different choices of $\tau$. The curves basically overlap in the later epochs because our steady updating rule is never broken in the later epochs.

## 5 CONCLUSION

In this paper, we revisited the training dynamics of the Transformers by visualizing the spectral norm of weight matrices, the activations and the attention map, presented a theoretical analysis for the Transformer training and identified two modes of attention entropy collapse, *i.e.*, the benign collapse and the malignant collapse, in which the malignant collapse accompanies model crash. Moreover, we revealed that the *spectral energy concentration* of $W_q^\top W_k$ is the reason behind the model crash, which causes the attention map to be sparse and simultaneously low-rank. Furthermore, we proposed a steady updating rule to resolve the problem of spectral energy concentration of $W_q^\top W_k$ by controlling the rapid growth of singular values, which can prevent the fast spectral energy concentration to a few directions and thus avoid the malignant entropy collapse. We conducted extensive experiments to verify the proposed strategy with ViT, Swin Transformer, and GPT, and demonstrated that the proposed strategy could effectively and stably train a model without using any learning rate warmup.

ETHICS STATEMENT

In this paper, we aim to provide a novel approach to train transformers without learning rate warmup. Our work does not involve any human subjects, and we have carefully ensured that it poses no potential risks or harms. Additionally, there are no conflicts of interest, sponsorship concerns, or issues related to discrimination, bias, or fairness associated with this study. We have taken steps to address privacy and security concerns, and all data used comply with legal and ethical standards. Our work fully adheres to research integrity principles, and no ethical concerns have arisen during the course of this study.

REPRODUCIBILITY STATEMENT

To ensure the reproducibility of our work, we provide all the details to reproduce the experiments. Theoretical proofs of the claims made in this paper, and detailed experimental settings and configurations are provided in the Appendices.

ACKNOWLEDGMENTS

Chun-Guang Li was partially supported by the National Natural Science Foundation of China under Grant 61876022. Qin Zou was partially funded by the National Natural Science Foundation of China under Grant 62171324.

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

## A  KRONECKER PRODUCT AND VECTORIZATION

Kronecker product (Graham, 2018; Petersen et al., 2008), also called as matrix direct product, is an operation defined on two matrices of arbitrary size. The specific definition is as follows.

**Definition 1 (Kronecker Product)**
*Let $A$ be an $n \times p$ matrix and $B$ an $m \times q$ matrix. The $mn \times pq$ matrix*

$$A \otimes B = \begin{bmatrix} a_{1,1}B & a_{1,2}B & \cdots & a_{1,p}B \\ a_{2,1}B & a_{2,2}B & \cdots & a_{2,p}B \\ \vdots & \vdots & \vdots & \vdots \\ a_{n,1}B & a_{n,2}B & \cdots & a_{n,p}B \end{bmatrix}$$

*is called the Kronecker product of $A$ and $B$. It is also called the direct product or the tensor product.*

For instance, if $A = \begin{bmatrix} 1 & 2 \\ 3 & 4 \end{bmatrix}$, and $B = \begin{bmatrix} 1 & 2 & 3 \\ 3 & 4 & 5 \end{bmatrix}$, then $A \otimes B = \begin{bmatrix} 1 & 2 & 3 & 2 & 4 & 6 \\ 3 & 4 & 5 & 6 & 8 & 10 \\ 3 & 6 & 9 & 4 & 8 & 12 \\ 9 & 12 & 15 & 12 & 16 & 20 \end{bmatrix}$.

Some basic properties of the Kronecker product include:

$$A \otimes (B \otimes C) = (A \otimes B) \otimes C,$$
$$A \otimes (B + C) = (A \otimes B) + (A \otimes C), \quad (A + B) \otimes C = (A \otimes C) + (B \otimes C),$$
$$(A \otimes B)^T = A^T \otimes B^T.$$

For a matrix $A$, the rank of $A \otimes A$ can be computed as,

$$\mathrm{rank}(A \otimes A) = \mathrm{rank}(A) \cdot \mathrm{rank}(A).$$

It means if the rank of the matrix $A$ is small, then the rank of $A \otimes A$ will also be very small.

In mathematics, *Vectorization* (Graham, 2018; Petersen et al., 2008) is usually used together with the Kronecker product to express matrix multiplication as a linear transformation on matrices. After vectorization, we can calculate the Jacobian matrix of the matrix product more conveniently. A property of vectorization for the matrix product is defined below.

**Proposition 2 (Property of Vectorization for Matrix Product)**
*Let $A \in \mathcal{R}^{m \times n}, B \in \mathcal{R}^{n \times k}, C \in \mathcal{R}^{k \times l}$, then we have*

$$\mathrm{vec}(ABC) = (C^\top \otimes A)\,\mathrm{vec}(B).$$

*Proof.* Let $C_i$ be the $i$-th row of $C$. Then we have:

$$\mathrm{vec}(ABC) = \sum_{i=1}^{n} \sum_{j=1}^{k} b_{ij}\,\mathrm{vec}(a_i C_j)$$

$$= \sum_{i=1}^{n} \sum_{j=1}^{k} b_{ij}(C_j^\top \otimes a_i)$$

$$= \sum_{j=1}^{k} (C_j^\top \otimes A) b_j$$

$$= (C^\top \otimes A)\,\mathrm{vec}(B).$$

$\square$

Furthermore, we have the following properties:

$$\mathrm{vec}(AB) = (I_k \otimes A)\,\mathrm{vec}(B) = (B^\top \otimes I_m)\,\mathrm{vec}(A),$$
$$\mathrm{vec}(ABC) = (C^\top B^\top \otimes I_m)\,\mathrm{vec}(A)$$
$$= (C^\top \otimes A)\,\mathrm{vec}(B)$$
$$= (I_l \otimes AB)\,\mathrm{vec}(C)$$

where $I_k \in \mathcal{R}^{k \times k}, I_l \in \mathcal{R}^{l \times l}, I_m \in \mathcal{R}^{m \times m}$ are all identity matrices.

Together with the Kronecker product, vectorization is an effective tool to compute matrix calculus. We can see the following two examples.

Let $P = AB$ where $A \in \mathcal{R}^{m \times n}, B \in \mathcal{R}^{n \times k}$, we have:

$$\frac{\partial \text{vec}(P)}{\partial \text{vec}(A)} = B^\top \otimes I_m, \quad \frac{\partial \text{vec}(P)}{\partial \text{vec}(B)} = I_k \otimes A.$$

Let $P = ABC$ where $A \in \mathcal{R}^{m \times n}, B \in \mathcal{R}^{n \times k}, C \in \mathcal{R}^{k \times l}$, we have,

$$\frac{\partial \text{vec}(P)}{\partial \text{vec}(A)} = C^\top B^\top \otimes I_m, \quad \frac{\partial \text{vec}(P)}{\partial \text{vec}(B)} = C^\top \otimes A, \quad \frac{\partial \text{vec}(P)}{\partial \text{vec}(C)} = I_l \otimes AB.$$

Vectorization and Kronecker product provide us a convenient way to analyze the self-attention module. We can compute the Jacobian matrix of the output with respect to the input or the weight matrix more conveniently. For more introduction to Kronecker product and vectorization, the readers can refer to (Petersen et al., 2008; Graham, 2018)

## B   DERIVATION OF JACOBIAN MATRIX FOR SINGLE-HEAD SELF-ATTENTION

A single-head self-attention can be defined as

$$Y = W_v X A,$$

where $A = \text{softmax}(\frac{P}{\sqrt{d_q}})$ in which $P = X^\top W_q^\top W_k X$. The matrix $A$ is called the attention matrix and $\frac{P}{\sqrt{d_q}}$ is called the logit, $A \in \mathcal{R}^{n \times n}, X \in \mathcal{R}^{d \times n}, W_v \in \mathcal{R}^{d_v \times d}$ . Here, our goal is to calculate $\frac{\partial \text{vec}(Y)}{\partial \text{vec}(X)}$.

In the main body, we have derived $\frac{\partial \text{vec}(P)}{\partial \text{vec}(X)}$. Here, let us calculate the matrix calculus of $A = \text{softmax}(\frac{P}{\sqrt{d_q}})$ with respect to $P$ using Kronecker products and vectorization. We can rewrite it as $A = \exp(\frac{P}{\sqrt{d_q}}) \oslash (1_n 1_n^\top \exp(\frac{P}{\sqrt{d_q}}))$, where $1_n$ denotes an $n$-dimensional vector of 1's in $\mathcal{R}^n$. Note that $A$ is obtained by conducting a softmax operation in each column individually.

First, let us define two intermediate variables:

$$B = \exp(\frac{P}{\sqrt{d_q}}), \quad C = 11^\top \exp(\frac{P}{\sqrt{d_q}}) = 11^\top B.$$

In this way, we can represent the attention matrix $A$ as $A = B \oslash C$.

Then, we can vectorize the equation $A = B \oslash C$ as follows:

$$\text{vec}(A) = \text{vec}(B \oslash C) = \text{vec}(B) \oslash \text{vec}(C).$$

According to the chain rule, we have

$$\frac{\partial \text{vec}(A)}{\partial \text{vec}(P)} = \frac{\partial \text{vec}(A)}{\partial \text{vec}(B)} \frac{\partial \text{vec}(B)}{\partial \text{vec}(P)} + \frac{\partial \text{vec}(A)}{\partial \text{vec}(C)} \frac{\partial \text{vec}(C)}{\partial \text{vec}(P)}.$$

Let us calculate each individual term. We have

$$\frac{\partial \text{vec}(A)}{\partial \text{vec}(B)} = \mathbf{1}_{n^2} \oslash \text{diag}(\text{vec}(C)),$$

$$\frac{\partial \text{vec}(B)}{\partial \text{vec}(P)} = \frac{\text{diag}(\text{vec}(B))}{\sqrt{d_q}},$$

$$\frac{\partial \text{vec}(A)}{\partial \text{vec}(C)} = -\text{diag}(\text{vec}(B) \oslash (\text{vec}(C) \odot \text{vec}(C))),$$

$$\frac{\partial \text{vec}(C)}{\partial \text{vec}(P)} = \frac{(I_n \otimes \mathbf{1}_n \mathbf{1}_n^\top)\text{diag}(\text{vec}(B))}{\sqrt{d_q}},$$

where $I_n$ is an identity matrix of $n \times n$ and $\otimes$ is the Kronecker product, $\mathbf{1}_{nn}$ denotes an $n^2$-dimensional vector of 1's in $\mathcal{R}^{n^2}$.

By substituting the above four terms into the chain rule, we have

$$\frac{\partial \text{vec}(A)}{\partial \text{vec}(P)} = \frac{\left(\mathbf{1}_{n^2} \oslash \text{diag}(\text{vec}(C))\right)\text{diag}(\text{vec}(B)) - \text{diag}(\text{vec}(B) \oslash (\text{vec}(C) \odot \text{vec}(C)))(I_n \otimes \mathbf{1}_n \mathbf{1}_n^\top)\text{diag}(\text{vec}(B))}{\sqrt{d_q}}$$

$$= \frac{\text{diag}(\text{vec}(A)) - \text{diag}(\text{vec}(B) \oslash (\text{vec}(C) \odot \text{vec}(C)))(I_n \otimes \mathbf{1}_n \mathbf{1}_n^\top)\text{diag}(\text{vec}(B))}{\sqrt{d_q}}$$

$$= \frac{\text{blockdiag}(\text{diag}(A_{:,1}) - A_{:,1}A_{:,1}^\top, \ldots, \text{diag}(A_{:,n}) - A_{:,n}A_{:,n}^\top)}{\sqrt{d_q}}.$$

For the simplicity, we denote

$$J = \text{blockdiag}(\text{diag}(A_{:,1}) - A_{:,1}A_{:,1}^\top, \ldots, \text{diag}(A_{:,n}) - A_{:,n}A_{:,n}^\top).$$

When $A$ and $P$ are vectorized into vectors, we use $a$ and $p$ to denote them, respectively. Then we see that

$$\frac{\partial \text{vec}(a)}{\partial \text{vec}(p)} = \frac{\text{diag}(a) - aa^\top}{\sqrt{d_q}}.$$

If $a$ approaches to a unit vector $e$, then the Jabobian matrix $\frac{\partial \text{vec}(a)}{\partial \text{vec}(p)}$ will tend to $\mathbf{0}$.

In Section 3.2, we have the following Jacobian matrix

$$\frac{\partial \text{vec}(P)}{\partial \text{vec}(X)} = (X^\top W_k^\top W_q \otimes I_n)K + (I_n \otimes X^\top W_q^\top W_k).$$

By vectorization of $Y = W_v X A$, we have

$$\partial \text{vec}(Y) = (A^\top \otimes W_v)\partial \text{vec}(X) + (I_n \otimes W_v X)\partial \text{vec}(A).$$

Therefore, according to the product rule and chain rule, we can denote the Jacobian matrix of $Y$ with respect to $X$ as follows:

$$\frac{\partial \text{vec}(Y)}{\partial \text{vec}(X)} = (A^\top \otimes W_v) + (I_n \otimes W_v X)\frac{\partial \text{vec}(A)}{\partial \text{vec}(X)},$$

$$= (A^\top \otimes W_v) + (I_n \otimes W_v X)\frac{\partial \text{vec}(A)}{\partial \text{vec}(P)}\frac{\partial \text{vec}(P)}{\partial \text{vec}(X)}.$$

Bringing in all these terms, we get the following formula:

$$\frac{\partial \text{vec}(Y)}{\partial \text{vec}(X)} = (A^\top \otimes W_v) + (I_n \otimes W_v X)\frac{J}{\sqrt{d_q}}\left((X^\top W_k^\top W_q \otimes I_n)K + (I_n \otimes X^\top W_q^\top W_k)\right). \qquad (7)$$

Let us analyze Equation 7. If a malignant entropy mode happens, $\frac{J}{\sqrt{d_q}}$ will approach $\mathbf{0}$ because each $\boldsymbol{a}$ in $\boldsymbol{A}$ will be a unit vector $\boldsymbol{e}$. From the perspective of the forward process, the features $\boldsymbol{Y}$ will collapse to several directions. From the perspective of the backward process, $\frac{J}{\sqrt{d_q}}$ will become $\mathbf{0}$, and $\frac{\partial \text{vec}(\boldsymbol{Y})}{\partial \text{vec}(\boldsymbol{X})}$ will be a sparse and simultaneously low-rank matrix. Through $\boldsymbol{A}^\top \otimes \boldsymbol{W}_v$, most of the positions in $\boldsymbol{X}$ will get zero gradient, and only very few columns will obtain some large noisy gradients. In a malignant entropy mode, the learned feature is invalid and useless. Similarly, if a benign entropy mode happens, the attention map $\boldsymbol{A}$ will approach an identity matrix $\boldsymbol{I}$ and $\frac{J}{\sqrt{d_q}} \approx 0$ when $\boldsymbol{A} \approx \boldsymbol{I}$. Therefore, we have $\frac{\partial \text{vec}(\boldsymbol{Y})}{\partial \text{vec}(\boldsymbol{X})} \approx (\boldsymbol{I}^\top \otimes \boldsymbol{W}_v)$. In this way, a self-attention module degenerates to a linear layer.

## C  PROOF OF BENIGN ENTROPY COLLAPSE

Recall that $\boldsymbol{A} = \text{softmax}(\frac{\boldsymbol{P}}{\sqrt{d_q}})$ where $\boldsymbol{P} = \boldsymbol{X}^\top \boldsymbol{W}_q^\top \boldsymbol{W}_k \boldsymbol{X}$. Here, let $\boldsymbol{W} = \boldsymbol{W}_q^\top \boldsymbol{W}_k$ and $\boldsymbol{W} \in \mathcal{R}^{d \times d}$. In this way, we have that $\boldsymbol{A} = \text{softmax}(\frac{\boldsymbol{X}^\top \boldsymbol{W} \boldsymbol{X}}{\sqrt{d_q}})$. We know that $\text{rank}(\boldsymbol{W}) \le d_q$ and $d_q < d$. To prove $\boldsymbol{A}$ will always collapse to an identity matrix when $\boldsymbol{W}$ is a non-symmetric positive quasi-definite square matrix, it is equivalent to prove $\mathbb{E}\left[\boldsymbol{x}_i^\top \boldsymbol{W} \boldsymbol{x}_i\right] \gg \mathbb{E}\left[\boldsymbol{x}_i^\top \boldsymbol{W} \boldsymbol{x}_j\right]$ for any $i \ne j$. It will be very hard to prove it mathematically if $\boldsymbol{W}$ is a form of a non-symmetric positive quasi-definite square matrix. Therefore, let us make some simplification assumptions. Assume $\boldsymbol{W}$ is a real symmetric positive semi-definite square matrix and its trace is in direct proportion to the dimension $d_q$, and any $\boldsymbol{x}_i$ is a high-dimension random vector and each element in $x_{i,j} \overset{\text{iid}}{\sim} \mathcal{N}(0,1)$.

We break our proof into two sub-problems.

**Proposition 3 (Expectation of $\boldsymbol{x}_i^\top \boldsymbol{W} \boldsymbol{x}_i$)**
*Let $\boldsymbol{W}$ be a real symmetric positive semi-definite matrix, and any $\boldsymbol{x}_i$ be a high-dimensional random vector. Then, we have $\mathbb{E}\left[\boldsymbol{x}_i^\top \boldsymbol{W} \boldsymbol{x}_i\right] = \text{trace}(\boldsymbol{W})$.*

*Proof.* Let $\boldsymbol{W}$ be a real symmetric positive semi-definite, thus it can be decomposed into $\boldsymbol{W} = \boldsymbol{U} \Sigma \boldsymbol{U}^\top$. In this way, we have

$$
\begin{aligned}
\mathbb{E}\left[\boldsymbol{x}_i^\top \boldsymbol{W} \boldsymbol{x}_i\right] &= \mathbb{E}\left[\boldsymbol{x}_i^\top \boldsymbol{U} \Sigma \boldsymbol{U}^\top \boldsymbol{x}_i\right] \\
&= \mathbb{E}\left[\boldsymbol{z}^\top \Sigma \boldsymbol{z}\right] \quad (\text{let } \boldsymbol{z} = \boldsymbol{U}^\top \boldsymbol{x}_i) \\
&= \mathbb{E}\left[\sum_{i=1}^{d} \sigma_i z_i^2\right] \quad (\Sigma \text{ is a diagonal matrix}) \\
&= \sum_{i=1}^{d_q} \sigma_i \times (0+1) \quad (\text{by independence, mean } 0) \\
&= \sum_{i=1}^{d_q} \sigma_i = \text{trace}(\boldsymbol{W}).
\end{aligned}
$$

For a real symmetric positive semi-definite, all its singular values are larger or equal to 0. Thus, we know $\sum_{i=1}^{d_q} \sigma_i > 0$ considering $\boldsymbol{W}$ is not a matrix of all zeros. $\square$

**Proposition 4 (Expectation of $\boldsymbol{x}_i^\top \boldsymbol{W} \boldsymbol{x}_j$ for $i \ne j$)**
*Let $\boldsymbol{W}$ be a real symmetric positive semi-definite square matrix, and any $\boldsymbol{x}_i$ is a high-dimension random vector, $\mathbb{E}_{i \ne j}\left[\boldsymbol{x}_i^\top \boldsymbol{W} \boldsymbol{x}_j\right] = 0$.*

*Proof.* Let $W$ be a real symmetric positive semi-definite. Thus, it can be decomposed into $W = U\Sigma U^\top$. In this way, we have

$$
\begin{aligned}
\mathbb{E}_{i \neq j}\left[\boldsymbol{x}_i^\top \boldsymbol{W} \boldsymbol{x}_j\right] &= \mathbb{E}_{i \neq j}\left[\boldsymbol{x}_i^\top \boldsymbol{U} \Sigma \boldsymbol{U}^\top \boldsymbol{x}_j\right] \\
&= \mathbb{E}\left[\boldsymbol{z}^\top \Sigma \boldsymbol{v}\right] \quad (\text{let } \boldsymbol{z} = \boldsymbol{U}^\top \boldsymbol{x}_i, \text{ and } \boldsymbol{v} = \boldsymbol{U}^\top \boldsymbol{x}_j) \\
&= \mathbb{E}\left[\sum_{i=1}^{d_q} \sigma_{ij} z_i v_j\right] \quad (\Sigma \text{ is a diagonal matrix. } \boldsymbol{z} \text{ and } \boldsymbol{v} \text{ are independent}) \\
&= 0.
\end{aligned}
$$

$\square$

According to Proposition 3 and Proposition 4, we can have that $\mathbb{E}\left[\boldsymbol{x}_i^\top \boldsymbol{W} \boldsymbol{x}_i\right] > \mathbb{E}\left[\boldsymbol{x}_i^\top \boldsymbol{W} \boldsymbol{x}_j\right]$ for any $i \neq j$. Considering that $W$ is usually a high-dimensional matrix and some of its singular values are significantly larger than 0. Thus, after the softmax operation, $A$ will always collapse to an identity matrix. In this way, the self-attention module degenerates into a linear projection module. The model fitting ability will decline, but model training will not crash. Our proof is based on matrix computations (Golub & Van Loan, 2013) and high-dimensional probability (Vershynin, 2018).

## D  PROOF OF MALIGNANT ENTROPY COLLAPSE

*Proof.* **Step 1**. To prove the sparsity of $A$, we must show that the number of non-zero elements in each column of $A$ is small.

Now, let's consider the properties of the matrix $P = X^T W X$. Since $X$ and $W$ are low-rank matrices, the matrix $P$ will also be low-rank. Specifically, the rank of $P$ is bounded by the rank of $X$ and $W$, which is much smaller than the dimension. In particular, $P$ has only a small number of significant singular values. This implies that the entries in $P$ are concentrated in a lower-dimensional subspace.

When we apply the softmax function to the rows of $P$, the function concentrates most of the probability mass on a few components of each column. This is because the softmax function is sharply peaked around the largest values in each row. The smaller values in each row contribute less to the sum in the denominator of the softmax function, and therefore, their corresponding entries in $A$ will be small.

Thus, in each column of $A$, only a small number of entries will be non-zero with high probability, and the rest will be close to zero. This establishes the sparsity of $A$.

**Step 2**. To prove the low-rankness of $A$, we turn to proving that $A$ is approximately low-rank.

As noted earlier, the matrix $P = X^T W X$ is low-rank. Specifically, the rank of $P$ is bounded by the ranks of $X$ and $W$, which are both small. Therefore, $P$ has only a few dominant singular values. Assume that the softmax function does not significantly change the rank of the matrix. The rank of $A$ is controlled by the rank of $P$, as the softmax operation only introduces nonlinearities that do not increase the rank.

Thus, since $P$ has a small number of dominant singular values, $A$, formed by the softmax of $P$, will also have a small number of significant singular values. This implies that $A$ is approximately low-rank.

In conclusion, under the assumptions that $X$ and $W$ are low-rank matrices with sufficiently large singular values, the matrix $A$ formed by applying the softmax function to $P$ will exhibit the properties of both sparsity and low-rankness with high probability.

$\square$

# E    PROOF OF WEYL'S INEQUALITY ON SINGULAR VALUES

Our derivation depends on Horn & Johnson (1991; 2012). Readers can refer to these material for more background information.

*Proof.* Before we prove Weyl's Inequality on singular values, let us review Courant-Fischer min-max principle that is important for analyzing the singular values of matrix.

**Theorem 3 (Courant-Fischer Min-max Principle for Singular Values)**
*Let $W \in \mathcal{R}^{m \times n}$ be a matrix where $m \geq n$. $W$ has ordered singular values $\sigma_1 \geq \sigma_2 \geq \cdots \geq \sigma_n \geq 0$. Then, for $i = 1, 2, \ldots, n$, we have*

$$\sigma_i(W) = \max_{\substack{S \subset \mathcal{R}^n \\ \dim(S)=i}} \min_{\substack{x \in S \\ \|x\|=1}} \|Wx\| = \min_{\substack{S' \subset \mathcal{R}^n \\ \dim(S')=n-i+1}} \max_{\substack{x \in S' \\ \|x\|=1}} \|Wx\|$$

*where the maximum is taken over all $i$-dimensional subspaces $S$ of $\mathcal{R}^n$, and the minimum is taken over all unit vectors $x$ in $S$.*

Let $W_1 = U\Sigma_1 V^\top$ and $W_2 = \mathcal{U}\Sigma_2 \mathcal{V}^\top$ be singular value decompositions of $W_1$ and $W_2$ with unitary matrix $V = [v_1, \ldots, v_n], \mathcal{V} = [\mathfrak{v}_1, \ldots, \mathfrak{v}_n]$ where $v_i, \mathfrak{v}_i \in \mathcal{R}^n$ and unitary matrix $U = [u_1, \ldots, u_m], \mathcal{U} = [\mathfrak{u}_1, \ldots, \mathfrak{u}_m]$, where $u_j, \mathfrak{u}_j \in \mathcal{R}^m$.

Let $i$ and $j$ be positive integers with $1 \leq i, j \leq n$ and $i + j \leq n + 1$. Let $S_1 \equiv \text{Span}\{v_i, \ldots, v_n\}$ and $S_2 \equiv \text{Span}\{\mathfrak{v}_j, \ldots, \mathfrak{v}_n\}$; notice that $\dim(S_1) = n - i + 1$ and $\dim(S_2) = n - j + 1$. Let $k \equiv \dim(S_1 \cap S_2)$, then we have

$$\dim(S_1 \cap S_2) = \dim(S_1) + \dim(S_2) - \dim(S_1 + S_2) = (n - i + 1) + (n - j + 1) - \dim(S_1 + S_2)$$
$$\geq (n - i + 1) + (n - j + 1) - n = n - (i + j - 1) + 1 \geq 1.$$

Because of the bounds assumed for $i$ and $j$. Thus, the subspace $S_1 \cap S_2$ has positive dimension $k$, $n - k + 1 \leq i + j - 1$, and we have

$$\sigma_{i+j-1}(W_1 + W_2) \leq \sigma_{n-k+1}(W_1 + W_2)$$
$$= \min_{\substack{S \subset \mathcal{R}^n \\ \dim(S)=k}} \max_{\substack{x \in S \\ \|x\|_2=1}} \|(W_1 + W_2)x\|_2$$
$$\leq \max_{\substack{x \in S_1 \cap S_2 \\ \|x\|_2=1}} \|(W_1 + W_2)x\|_2$$
$$\leq \max_{\substack{x \in S_1 \cap S_2 \\ \|x\|_2=1}} \|W_1 x\|_2 + \max_{\substack{x \in S_1 \cap S_2 \\ \|x\|_2=1}} \|W_2 x\|_2$$
$$\leq \max_{\substack{x \in S_1 \\ \|x\|_2=1}} \|W_1 x\|_2 + \max_{\substack{x \in S_2 \\ \|x\|_2=1}} \|W_2 x\|_2 = \sigma_i(W_1) + \sigma_j(W_2).$$

As a special case, the second part of the theorem follows directly from the general result of part (a). Specifically, for $i = j = 1$, we have:

$$\sigma_1(W_1 + W_2) \leq \sigma_1(W_1) + \sigma_1(W_2).$$

This completes the proof.                                                                      □

## F   RELATED WORKS

**Training Dynamics of Transformer.** Previous works have delved into understanding the training dynamics of Transformers from two different perspectives: a high-level perspective and a low-level perspective. From a high-level perspective, Scan&Snap (Tian et al., 2023a) unveiled complex phenomena, particularly in single-layer architectures, relating to frequency and discriminative bias. These studies linked sparse attention patterns to token co-occurrence frequencies and observed two-stage behaviors in attention logits. JoMA (Tian et al., 2023b) further improved upon previous models by incorporating residual connections and MLP nonlinearity, analyzing joint training of MLP and self-attention layers, and offering qualitative explanations for multi-layer Transformer dynamics. From a low-level perspective, two critical challenges in Transformer training have been identified: rank collapse (Dong et al., 2021; Noci et al., 2022), where attention output converges to a rank 1 matrix, potentially causing vanishing gradients; and entropy collapse (Zhai et al., 2023), which denotes pathological low attention entropy, corresponding to highly concentrated attention scores. *In this work, we analyze and prove two different entropy collapse modes and identify the key reason for model failure is spectral energy concentration. Finally, we introduce a simple but effective solution to address this problem.*

**Training Stability of Transformer.** ReZero (Bachlechner et al., 2021) introduces a simple yet effective mechanism for improving training stability. The key innovation lies in initializing residual connections to zero, which allows networks to learn identity mappings more easily. Admin (Liu et al., 2020) introduces a new network initialization strategy tailored for Transformers to make the network train stable. DeepNorm (Wang et al., 2022) extends the concept of normalization to accommodate increasingly deeper networks. By dynamically adjusting normalization parameters, DeepNorm ensures stability even as network depth increases. LipsFormer Qi et al. (2023a) addresses the specific challenge of stability in transformer networks. By introducing a Lipschitz continuity constraint, Lipsformer effectively mitigates the issue of exploding gradients - a common problem in deep transformer architectures. This approach ensures that the network's output changes smoothly with respect to its input, promoting overall stability. ReZero, Admin, and DeepNorm can all be considered as an approach to control the Lipschitz constant of the network in the initial stage. *In this work, by revisiting the training dynamics of Transformer, we can achieve a stable training only by modifying the optimizer instead of using learning rate warmup or changing the network structures as LipsFormer (Qi et al., 2023a) and QKNorm Henry et al. (2020); Dehghani et al. (2023).*

**Learning Rate Schedule.** Warmup (Loshchilov & Hutter, 2016) has emerged as a must-have technique for ensuring a stable network training, especially in the initial phases of the optimization process. This method involves gradually increasing the learning rate from a small value to the desired initial learning rate over a certain number of training steps or epochs. The cosine learning rate scheduler (Loshchilov & Hutter, 2016) has gained popularity due to its smooth annealing properties. This schedule decreases the learning rate following a cosine curve, starting from an initial value and decaying to a minimum value over a set number of epochs or iterations. Cyclic learning rates (Smith, 2017) involve systematically varying the learning rate between boundary values. The learning rate oscillates between a lower and upper bound, either linearly or following other patterns (e.g., triangular, cosine). The above-mentioned learning rate schedules require specification of a stopping time step $T$, Defazio et al. (2024) introduces a Schedule-Free approach that avoids the need for this stopping time by eschewing the use of schedules entirely.

Compared to the up-mentioned works, the core novel contributions of our work lie on follows.

1. We present a theoretical analysis for Transformer training and point out two entropy collapse modes, *i.e.* the benign collapse and the malignant collapse.

2. We reveal that *spectral energy concentration (SEC) of* $W_q{}^\top W_k$ is the main reason of model crash.

3. We introduce AdamW$^2$, a new optimization strategy motivated by Weyl's Inequality.

We also observe there are two works (Kosson et al., 2023; 2024) discussing the needs of learning rate warmup by explicitly controlling the angular updates via Rotational Optimizer Variants and by limiting the Frobenius norm of the update relative to that of the weights. They provided some different perspectives on the necessity of the learning rate warmup.

## G    SIMULATION OF THREE ATTENTION MODES

We provide a simple simulation code to simulate three attention modes, but it is important to note that the real picture is more complicated. In real case, in the benign attention entropy mode, $W_q^\top W$ is a non-symmetric positive quasi-definite square matrix instead of a symmetric positive definite matrix in our simulation. The code is just to demonstrate the core ideas behind three attention modes.

CODE 1: Simulation of Three Attention Modes.

```python
import torch
import torch.nn
import matplotlib.pyplot as plt
import numpy as np

#Randomly generate data and weight matrices
d_q, d, num_tokens= 64, 768, 197
Wq = torch.randn(d_q, d)
Wk = torch.randn(d_q, d)
X = torch.randn(d, num_tokens)
W = torch.mm(Wq.T, Wk)

# Normal attention mode
W1 = W
P = torch.mm(torch.mm(X.T, W1), X)
attn_map1 = P.softmax(dim=1)

# Malignant attention entropy collapse mode
u,s,v = torch.svd(W)
s[0:3] = torch.tensor([3., 2., 1.])*s[0:3]
s[3:]  = 0.0
W2 = torch.mm(torch.mm(u, torch.diag(s)), v.T)
P = torch.mm(torch.mm(X.T, W2), X)
attn_map2 = P.softmax(dim=1)

# Benign attention entropy collapse mode
u,s,v = torch.svd(W)
W3 = torch.mm(torch.mm(u, torch.diag(s)), u.T)
P = torch.mm(torch.mm(X.T, W3), X)
attn_map3 = P.softmax(dim=1)

# Plot figures
fig, axs = plt.subplots(1, 3, figsize=(15, 5))
axs[0].imshow(attn_map1.detach().numpy())
axs[1].imshow(attn_map2.detach().numpy())
axs[2].imshow(attn_map3.detach().numpy())
plt.show()
```

## H   ATTENTION MAP VISUALIZATION OF GPT

Figure 7 visualizes the dynamic process of attention map as the number of training steps increases for a successful and unsuccessful GPT-Small model. It should be noted that the GPT model uses a lower triangular attention mask.

(a) *Block 11 (successful).*          (b) *Block 11 (unsuccessful).*

FIGURE 7: Visualization of the dynamic process of attention map as the number of training steps increases for a successful and unsuccessful GPT-Small model. Attention map gradually becomes sparse and low-rank along with the training process in a failure case. *Please click the images to play the flash. Best viewed with Acrobat Reader.*

In Figure 7, the attention values in a successful case distribute to different position, but the attention values in a unsuccessful case will only concentrate into several directions.

## I   MORE TRAINING DYNAMICS OF VIT AND GPT

Figure 8 visualizes a successful ViT training process. Compared with Figure 1, we find several significant differences as follows.

- In a successful ViT training process, the value of $\sigma_1(W_q^\top W_k)$ increases to 16,000, then starts to oscillate smoothly. But for an unsuccessful training, the value suddenly increases to a very large value, around 300,000, it triggers the model crash,
- The $\gamma_1$ and $\beta_1$ in a successful ViT training process are very smooth, but they change a lot in an unsuccessful case,
- The fast increase of $\sigma_1(W_q^\top W_k)$ is accompanied by a fast increase of $W_q$ and $W_k$.

We can observe similar phenomenon in Figure 9 and Figure 15. In a successful GPT training process, the value of $\sigma_1(W_q^\top W_k)$ increases to 60, then starts to oscillate smoothly. But for an unsuccessful GPT training, the value increases to 20,000. The difference between the sclae of value between GPT and ViT may be due to the density and sparsity of the supervision signal. In GPT, each token will contribute a gradient, but in ViT, only one class label in an image provides a supervision information.

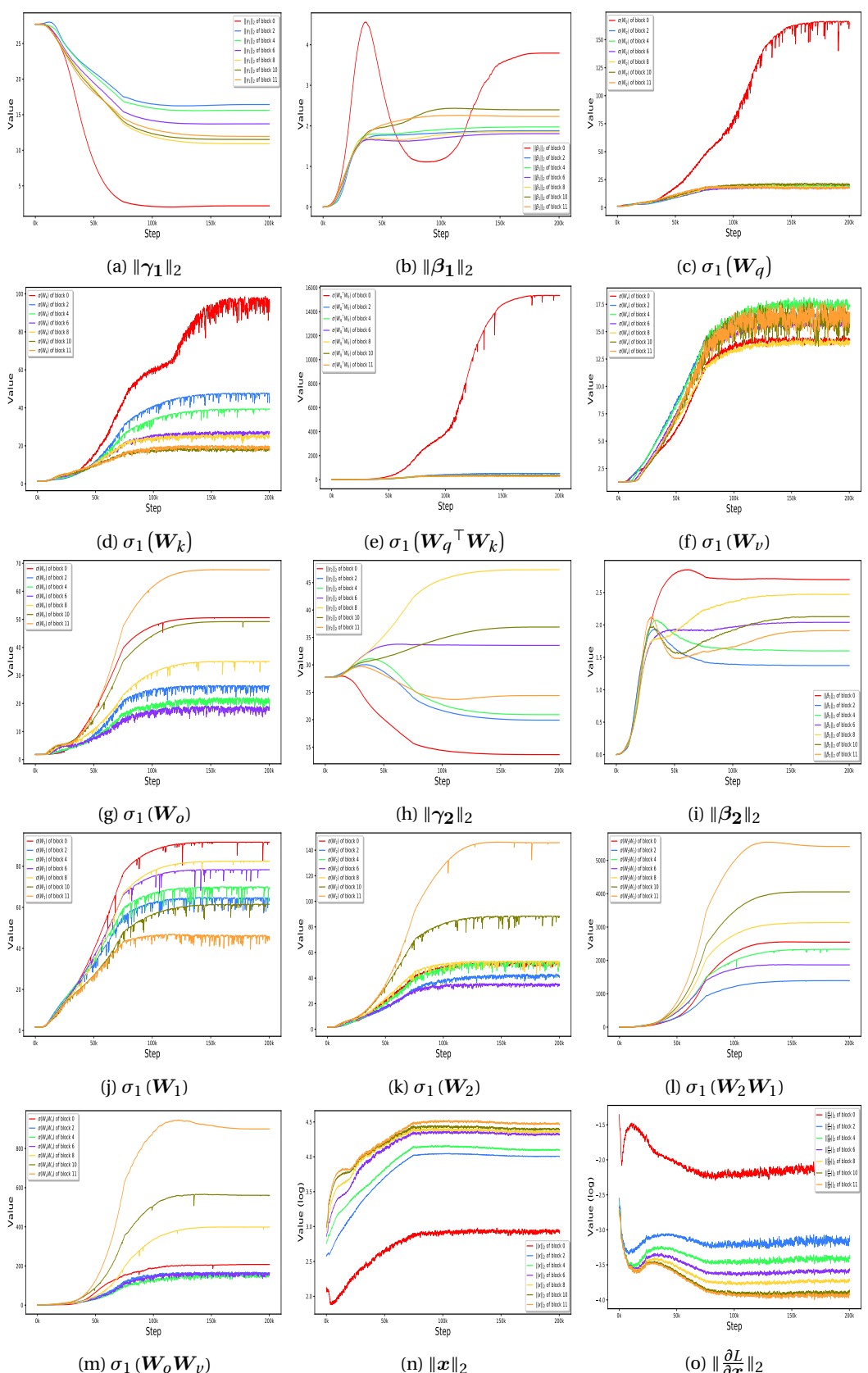

FIGURE 8: Training dynamics of a successful ViT training.

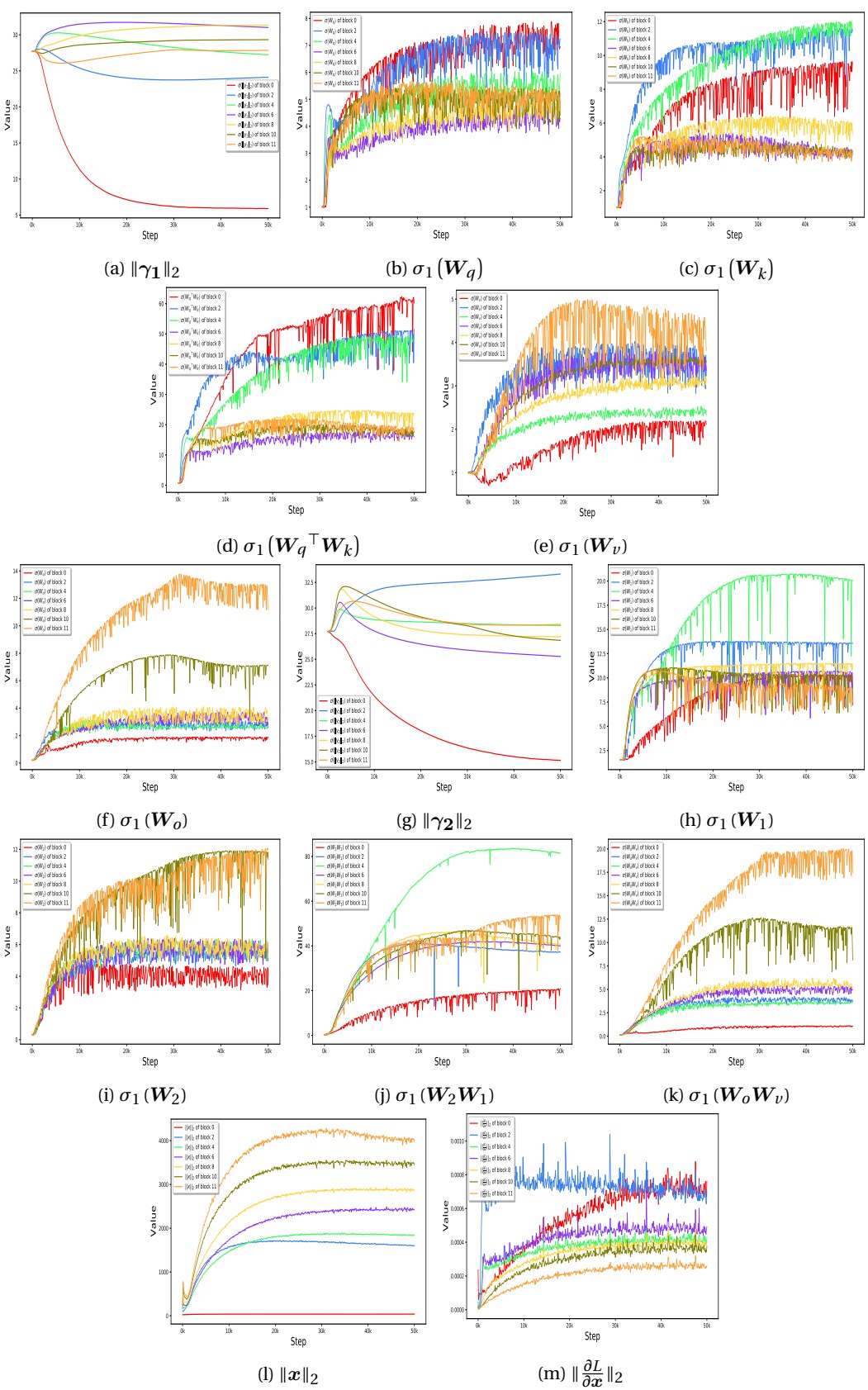

FIGURE 9: Training dynamics of a successful GPT training.

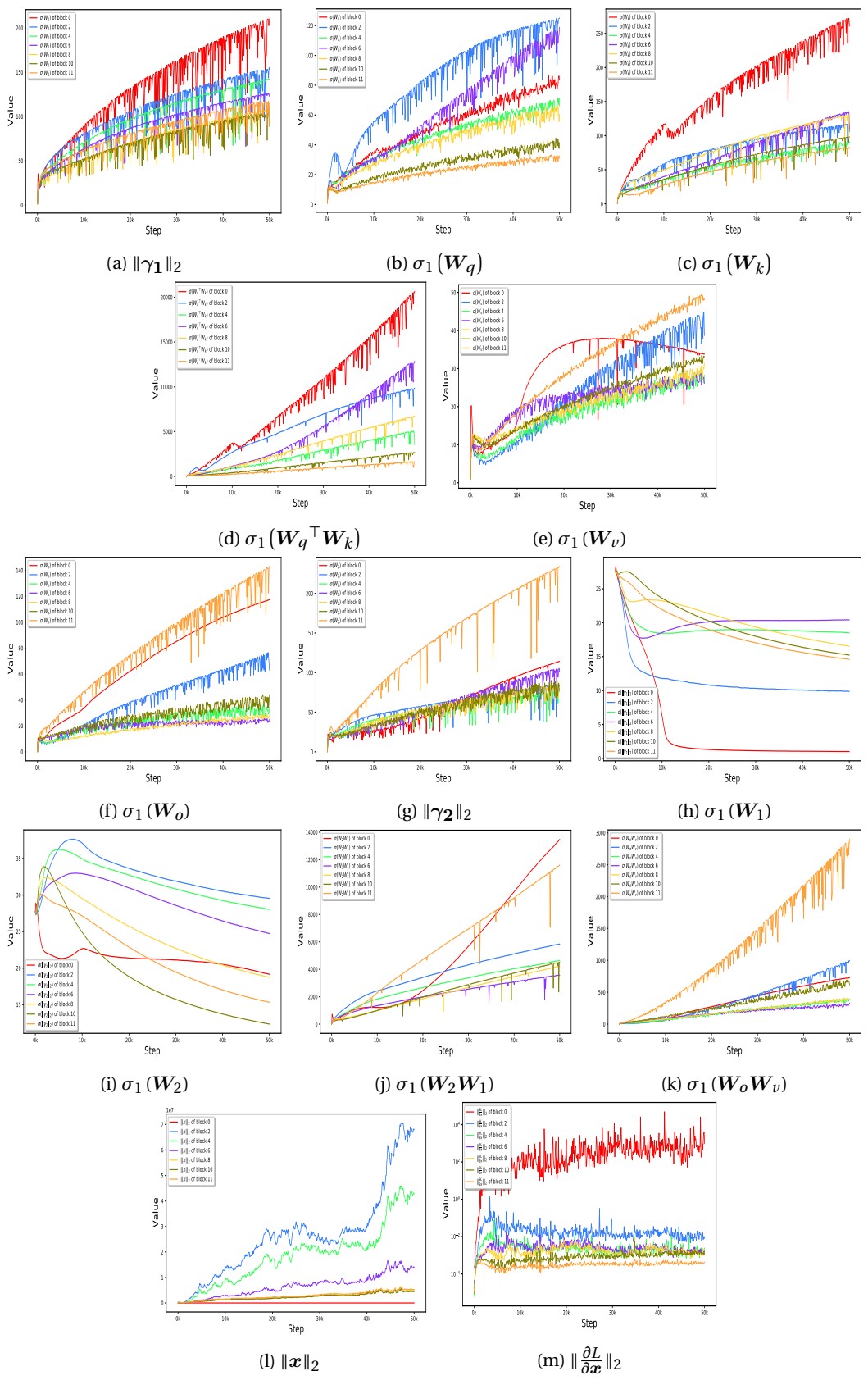

FIGURE 10: Training dynamics of an unsuccessful GPT training.

## J EXPERIMENT OF 1B VIT

To further evaluate the effectiveness of our method at a larger scale, we assessed ViT-g with 1B parameters. The ViT-g model architecture consists of 40 layers with a hidden dimension of 1408, 16 attention heads, and an MLP dimension of 6144. The total parameter count is 1011M, around one billion parameters. We conducted a comparative study between ViT-g with AdamW$^2$ and ViT-g with AdamW, where ViT-g with AdamW was evaluated under two settings: with and without learning rate warmup. Our AdamW$^2$ does not use warmup. The comparison results are presented in Figure 11 and Figure 12.

Figure 11 shows that ViT-g with AdamW crashes after only a few training steps when running without warmup. While the use of warmup enables ViT-g to complete training, but the loss spikes one time. Our ViT-g with AdamW$^2$ not only achieves stable training without warmup but also demonstrates better performance.

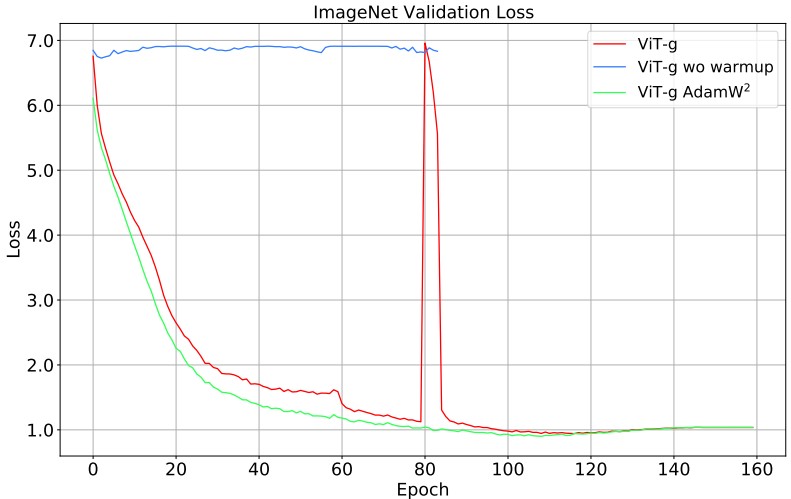

FIGURE 11: Comparison of loss curve of AdamW$^2$ and AdamW on ViT-g model.

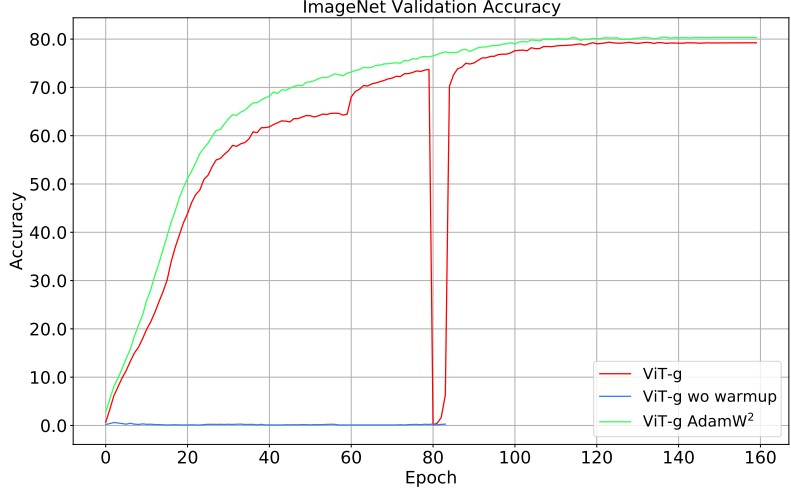

FIGURE 12: Comparison of accuracy of AdamW$^2$ and AdamW on ViT-g model.

## K   EXPERIMENT OF 774M NANOGPT

We also evaluated the effectiveness of our method on a larger-scale language model, termed as nanoGPT-large. The model architecture consists of 36 layers with a hidden dimension of 1280 and 20 attention heads. The total parameter count is 774M. Our experimental setup strictly follows the nanoGPT configuration, including all learning rate settings. It is important to note that training nanoGPT-large is computationally intensive, requiring two weeks to train 600K steps on 16 A800 GPUs. To reduce the training time, we limited our training to 100K steps instead of the full 600K steps. The comparison results are presented in Figure 13. We can see from Figure 13, nanoGPT-large achieves a stable training without warmup and obtains a similar validation loss with its counterpart, GPT2-large. This further verifies our understanding to the model crash of Transformer.

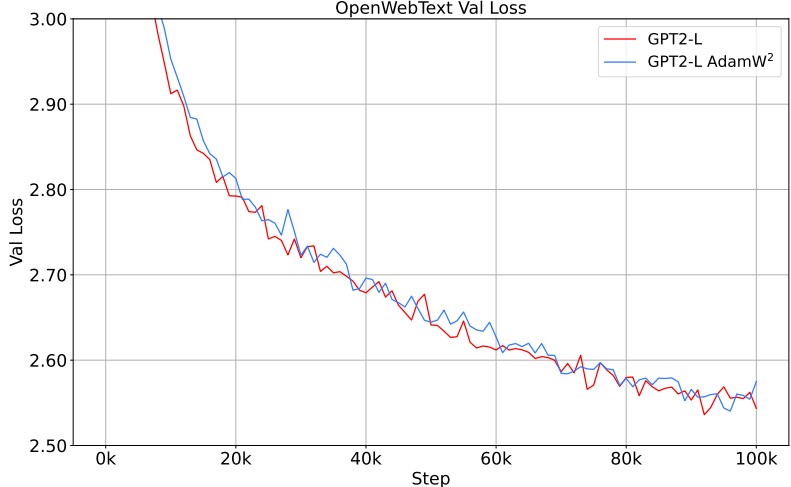

FIGURE 13: Comparison of validation loss of AdamW$^2$ and AdamW on nanoGPT-large model.

## L   EXPERIMENT OF FLATTEN-SWIN

Besides ViT, GPT, and Swin-Transformer, we further validated our approach on Flatten-Transformer Han et al. (2023). We used Flatten-Swin, and we compared the performance of our method and the baseline method training 150 and 300 epochs. Our method could stably train and demonstrate performance comparable to the baseline. This further verified the correctness of our understanding of neural network stability.

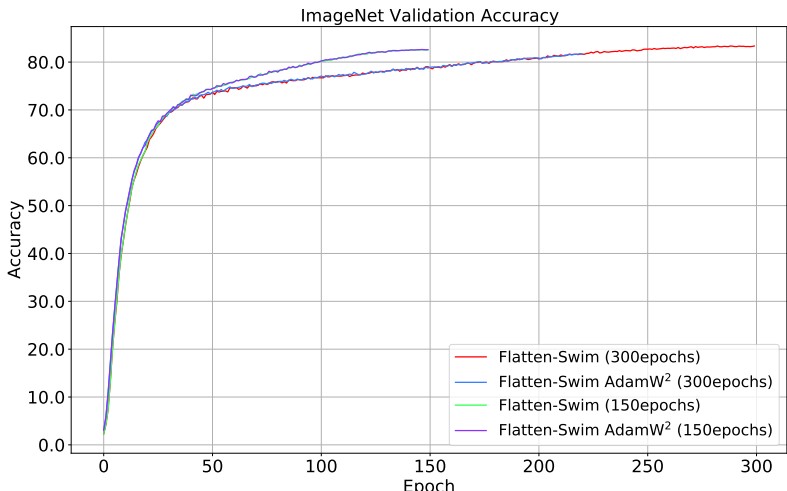

FIGURE 14: Evaluation of Flatten-Swin.

# M ACTUAL LEARNING RATE CURVE ALONG WITH TRAINING STEPS

We recorded the actual learning rate throughout the training steps, we sample one point every 50 steps. Our initial setting of the learning rate is a cosine learning rate scheduler without warmup. If $\alpha_t \frac{\sigma_1(\nabla W_t)}{\sigma_1(W_{t-1})} > \tau$, then $\alpha_t$ will be truncated to $\tau \frac{\sigma_1(W_{t-1})}{\sigma_1(\nabla W_t)}$. From the figure, we observe that $\gamma_1$ and $\gamma_2$ in RMSNorm only exceed the preset $\tau$ during the initial training phase and rarely exceed it afterwards. For other curves, they somewhat look like a curve with learning rate warmup, but we can see that different blocks have different learning rates.

We also observe that shallower layers are more likely to violate the preset $\tau$ value. It means the shallower layers are more likely to lead to a greater update of weight matrix and typically require a smaller learning rate. Additionally, we notice that for the weight matrix $W_2$, it is more prone to exceeding the preset $\tau$ value compared to the weight matrix $W_1$.

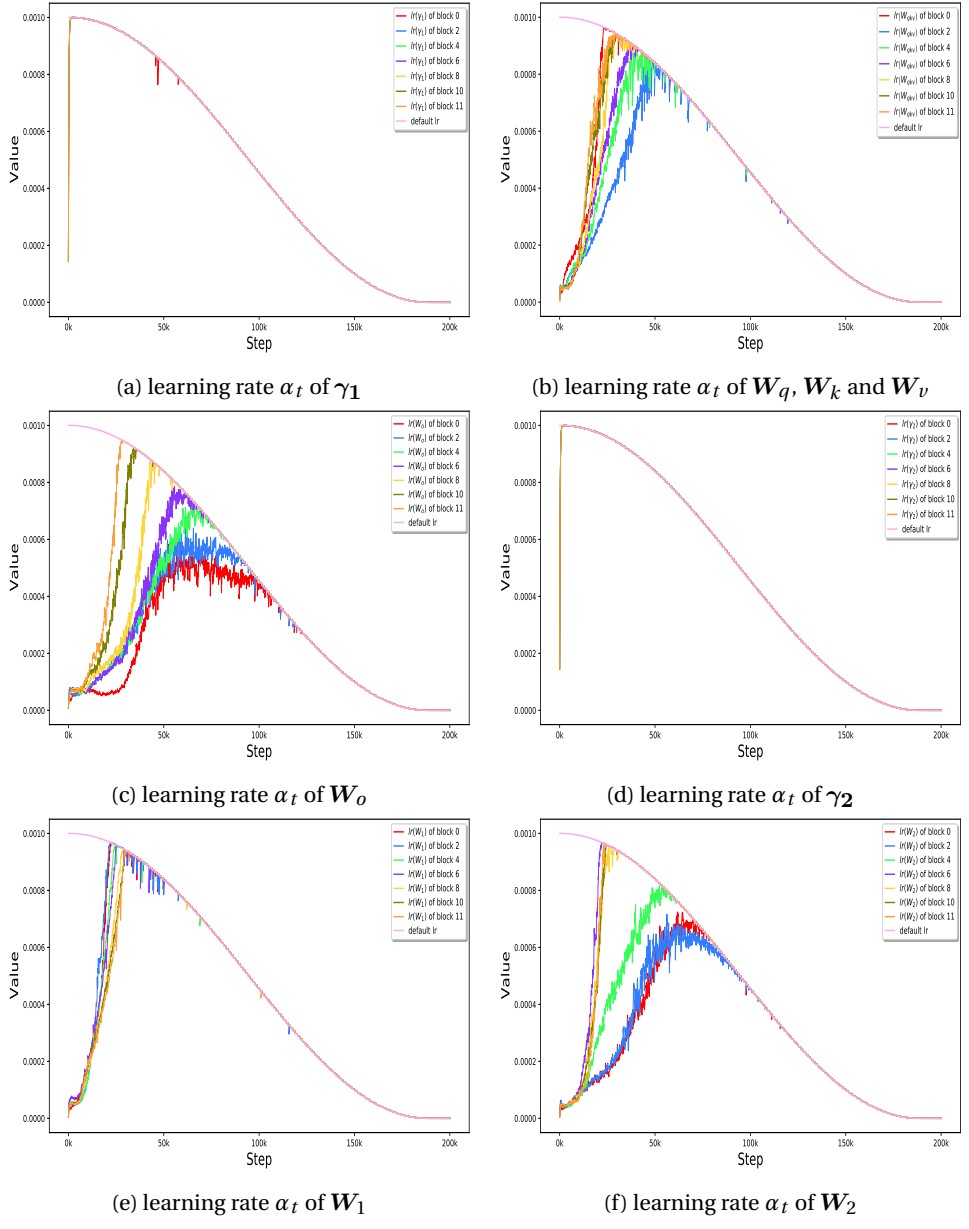

(a) learning rate $\alpha_t$ of $\gamma_1$

(b) learning rate $\alpha_t$ of $W_q$, $W_k$ and $W_v$

(c) learning rate $\alpha_t$ of $W_o$

(d) learning rate $\alpha_t$ of $\gamma_2$

(e) learning rate $\alpha_t$ of $W_1$

(f) learning rate $\alpha_t$ of $W_2$

FIGURE 15: Actual Learning Rate Curve along with Training steps.

## N    TRAINING CONFIGURATIONS

**Training Configurations.** We list the training configurations of ViT, GPT, Swin-Transformer and Flatten-Swin in Table 2. For ViT, GPT, Swin-Transformer and Flatten-Swin, we do not use learning rate warmup. For GPT, we follow the experimental configurations of nanoGPT (Karpathy, 2022), all parameters are same as GPT2 (Radford et al., 2019). For ViT, we use Timm (Wightman, 2019). For Swin-Transformer, we use the original code provided by Liu et al. (2021). For Flatten-Swin, we use the original code provided by (Han et al., 2023).

TABLE 2: Training configurations for ViT, GPT and Swin-Transformer.

(a) Training configurations for ViT.

| training config | ViT-B/L/g ($224^2$) |
|---|---|
| optimizer | AdamW$^2$ |
| $\tau$ (In default) | 0.004 or 0.003 |
| warmup epochs | 0 |
| weight init | Truncated Xavier |
| base learning rate | 1e-3 |
| weight decay | 0.05/0.1 |
| optimizer momentum | $\beta_1, \beta_2 = 0.9, 0.99$ |
| batch size | 1024 |
| training epochs | 150 |
| learning rate schedule | cosine decay |
| randaugment | $(9, 0.5)$ |
| mixup | 0.8 |
| cutmix | 1.0 |
| random erasing | 0 |
| label smoothing | 0.1 |
| stochastic depth | 0.1/0.5 |
| gradient clip | None |
| exp. mov. avg. (EMA) | no |

(b) Training configurations for GPT.

| training config | GPT-S/L |
|---|---|
| optimizer | AdamW$^2$ |
| $\tau$ | 0.01 |
| warmup epochs | 0 |
| weight init | Xavier |
| baseline learning rate | 0.0006 or 0.00025 |
| weight decay | 0.1 |
| optimizer momentum | $\beta_1, \beta_2 = 0.9, 0.95$ |
| tokens seen each update | 500,000 |
| max iters | 600K or 100K |
| batch size | 480 |
| sequence length | 1024 |
| dropout | 0.0 |
| bfloat16 | True |
| gradient clipping | 1.0 |

(c) Training configurations for Swin-Transformer.

| training config | Swin S/B ($224^2$) |
|---|---|
| optimizer | AdamW$^2$ |
| $\tau$ (In default) | 0.004 |
| warmup epochs | 0 |
| training epochs | 300 |
| others | same as Liu et al. (2021) |

(d) Training configurations for Flatten-Swin.

| training config | Flatten-Swin S ($224^2$) |
|---|---|
| optimizer | AdamW$^2$ |
| $\tau$ (In default) | 0.004 |
| warmup epochs | 0 |
| training epochs | 150 or 300 |
| others | same as Han et al. (2023) |

## O    NON-SYMMETRIC POSITIVE QUASI-DEFINITE SQUARE MATRIX

When we mention a non-symmetric positive quasi-definite square matrix, we mean it has the following three properties,

1. $\boldsymbol{W}_q^\top \boldsymbol{W}_k$ is not symmetric because generally, $\boldsymbol{W}_q^\top \boldsymbol{W}_k \neq \boldsymbol{W}_k^\top \boldsymbol{W}_q$,

2. $\boldsymbol{W}_q^\top \boldsymbol{W}_k$ is a square matrix and most of its eigenvalues are larger than 0, and only very few are less than 0.0. So we call it positive quasi-definite matrix.

3. if we assume $\boldsymbol{W} = \boldsymbol{W}_q^\top \boldsymbol{W}_k$ is positive definite matrix, if for each element in $\boldsymbol{x}$ is sampled from a standard Gaussian distribution, we can prove

$$\mathbb{E}\left[\boldsymbol{x_i}^\top \boldsymbol{W} \boldsymbol{x_i}\right] \gg \mathbb{E}\left[\boldsymbol{x_i}^\top \boldsymbol{W} \boldsymbol{x_j}\right]$$

when $i \neq j$, see Appendix C for the proof.

## P  DISCUSSION ABOUT RANK COLLAPSE, ENTROPY COLLAPSE AND SPARSE YET LOW-RANK ENTROPY MATRIX

Before we start our discussion, let us see three matrices,

$$A = \begin{pmatrix} \frac{1}{5} & \frac{1}{5} & \frac{1}{5} & \frac{1}{5} & \frac{1}{5} \\ \frac{1}{5} & \frac{1}{5} & \frac{1}{5} & \frac{1}{5} & \frac{1}{5} \\ \frac{1}{5} & \frac{1}{5} & \frac{1}{5} & \frac{1}{5} & \frac{1}{5} \\ \frac{1}{5} & \frac{1}{5} & \frac{1}{5} & \frac{1}{5} & \frac{1}{5} \\ \frac{1}{5} & \frac{1}{5} & \frac{1}{5} & \frac{1}{5} & \frac{1}{5} \end{pmatrix}, B = \begin{pmatrix} 1 & 0 & 0 & 0 & 0 \\ 0 & 1 & 0 & 0 & 0 \\ 0 & 0 & 1 & 0 & 0 \\ 0 & 0 & 0 & 1 & 0 \\ 0 & 0 & 0 & 0 & 1 \end{pmatrix}, C = \begin{pmatrix} 1 & 0 & 0 & 0 & 0 \\ 1 & 0 & 0 & 0 & 0 \\ 1 & 0 & 0 & 0 & 0 \\ 1 & 0 & 0 & 0 & 0 \\ 1 & 0 & 0 & 0 & 0 \end{pmatrix}$$

We can see that $A$ is low-rank, $B$ is sparse but not low-rank, $C$ is sparse and low-rank.

In previous papers (Dong et al., 2021; Zhai et al., 2023), researchers have analyzed the problem of model crash via *rank collapse of activations and entropy collapse of attention map*. Dong et al. (Dong et al., 2021) attributes the model crash into *rank collapse of the activations*, but Zhai et al. (2023) think it is the *entropy collapse of the attention map* leading to the model crash.

However, based on our analysis, we can find a counterexamples for entropy collapse, and meanwhile the rank collapse of the activation cannot fully describe the inner reason of the model crash (the weight matrix instead of activations). When the state of $C$ usually happens, the model crashed,

- Rank collapse of the activations cannot reveal the underlying cause that exists in the weight matrix. Weight matrix is the inner key ingredient of the model instead of activations.

- $B$ is a counterexample of entropy collapse. we observe that in some successful cases, state $B$ occurs. According to the definition of entropy collapse, state $B$ should lead to model crash; however, our experiments show that the model remains stable in this state.

- Sparse yet low-rank attention matrix is the state of the attention map when a model crashs. We believe rank collapse of activations and entropy collapse of attention map are not enough to describe the state of the model crash precisely. According to our analysis, the Spectral Energy Concentration (SEC) of the $W_q^\top W_k$ is the inner reason the model crash, and the sparse yet low-rank attention matrix is the phenomena observed on the attention matrix.

In summary, our paper, via a rigid theoritical analysis, our paper reveals the Spectral Energy Concentration (SEC) of the $W_q^\top W_k$ is the inner reason the model crash, and the sparse yet low-rank attention matrix is the phenomena that is observed on the attention matrix.

## Q  ILLUSTRATION OF FIGURE 5

**Illustration of Arrow 1.**

According to the property of Kronecker Product, we have

$$\text{rank}(X \otimes X) = \text{rank}(X) \cdot \text{rank}(X).$$

Since $X$ is low-rank, then $X \otimes X$ is also low-rank. In the following, we will also prove the singular values of $X \otimes X$ will also strengthen the concertration of spectral energy into some directions with large singular values.

**Illustration of Arrow 2.**

According to the computation of Jacobian matrix, we have

$$\frac{\partial \operatorname{vec}(\boldsymbol{P})}{\partial \operatorname{vec}(\boldsymbol{W_q}^\top \boldsymbol{W_k})} = \boldsymbol{X}^\top \otimes \boldsymbol{X}^\top$$

where $\boldsymbol{P} = \boldsymbol{X}^\top \boldsymbol{W}_q^\top \boldsymbol{W}_k \boldsymbol{X}^\top$.

**Illustration of Arrow 3.**

Let $\boldsymbol{X} \in \mathbb{R}^{m \times n}$ be a matrix with rank $r \le \min(m, n)$. Denote the singular values of $\boldsymbol{X}$ as $\sigma_1 \ge \sigma_2 \ge \cdots \ge \sigma_r > 0$ and $\sigma_{r+1} = \cdots = \sigma_{\min(m,n)} = 0$.

Definition. (Singular Values of Kronecker Product)** For a matrix $\boldsymbol{X}$, define $\Lambda(\boldsymbol{X})$ as the set of all possible products of its singular values, i.e., $\Lambda(\boldsymbol{X}) = \{\sigma_i \sigma_j : 1 \le i, j \le r\}$.

Theorem (Singular Values of Kronecker Product) For a low-rank matrix $\boldsymbol{X} \in \mathbb{R}^{m \times n}$ with rank $r$, the singular values of $\boldsymbol{X} \otimes \boldsymbol{X}$ are precisely the elements in $\Lambda(\boldsymbol{X})$. More formally, let $\{\mu_k\}$ be the set of singular values of $\boldsymbol{X} \otimes \boldsymbol{X}$. Then $\{\mu_k\} = \{\sigma_i \sigma_j \text{ where} 1 \le i, j \le r\}$.

Proof. Consider the singular value decomposition (SVD) of $\boldsymbol{X} = \boldsymbol{U} \boldsymbol{\Sigma} \boldsymbol{V}^T$, where $\boldsymbol{U} \in \mathbb{R}^{m \times m}$ is an orthogonal matrix, $\boldsymbol{V} \in \mathbb{R}^{n \times n}$ is an orthogonal matrix, and $\boldsymbol{\Sigma} = \operatorname{diag}(\sigma_1, \ldots, \sigma_r, 0, \ldots, 0)$. Then, the Kronecker product $\boldsymbol{X} \otimes \boldsymbol{X}$ can be expanded as: $\boldsymbol{X} \otimes \boldsymbol{X} = (\boldsymbol{U} \boldsymbol{\Sigma} \boldsymbol{V}^T) \otimes (\boldsymbol{U} \boldsymbol{\Sigma} \boldsymbol{V}^T) = (\boldsymbol{U} \otimes \boldsymbol{U})(\boldsymbol{\Sigma} \otimes \boldsymbol{\Sigma})(\boldsymbol{V}^T \otimes \boldsymbol{V}^T)$. The singular values of $\boldsymbol{U} \otimes \boldsymbol{U}$ and $\boldsymbol{V}^T \otimes \boldsymbol{V}^T$ are all 1, as they are composed of orthogonal matrices.

Therefore, the singular values of $\boldsymbol{X} \otimes \boldsymbol{X}$ are precisely the elements of $\boldsymbol{\Sigma} \otimes \boldsymbol{\Sigma}$, which are exactly $\{\sigma_i \sigma_j : 1 \le i, j \le r\}$.

In summary, the Singular Values of Kronecker Product $\boldsymbol{X} \otimes \boldsymbol{X}$ will also strengthen the concertration of spectral energy into some directions with large singular values. In this way, the singular values of $\frac{\partial \operatorname{vec}(\boldsymbol{P})}{\partial \operatorname{vec}(\boldsymbol{W_q}^\top \boldsymbol{W_k})}$ will also concentrate to a few directions.

When we update $\boldsymbol{W}_q^\top \boldsymbol{W}_k$ according to the following equation:

$$\operatorname{vec}(\boldsymbol{W}_q^\top \boldsymbol{W}_k)_{new} = \operatorname{vec}(\boldsymbol{W}_q^\top \boldsymbol{W}_k)_{old} - \alpha \frac{\partial \mathcal{L}}{\partial \operatorname{vec}(\boldsymbol{P})} \frac{\partial \operatorname{vec}(\boldsymbol{P})}{\partial \operatorname{vec}(\boldsymbol{W}_q^\top \boldsymbol{W}_k)}$$

where $\mathcal{L}$ is the loss function, and $\alpha$ is the step size. Since the singular values of $\frac{\partial \operatorname{vec}(\boldsymbol{P})}{\partial \operatorname{vec}(\boldsymbol{W_q}^\top \boldsymbol{W_k})}$ will also concentrate to a few directions, the update will lead to the singular values of $\boldsymbol{W}_q^\top \boldsymbol{W}_k$ tends to concentrate.

**Illustration of Arrow 4.**

Our Theorem 1 in the paper is to demonstrate that the spectral energy concentration of $\boldsymbol{W}_q^\top \boldsymbol{W}_k$ and the associated dominant large singular values will lead to $\boldsymbol{A}$ to be a sparse yet low-rank matrix.

**Illustration of Arrow 5.**

Since $\boldsymbol{X}^{l+1} = \boldsymbol{V} \boldsymbol{A}$, where $\boldsymbol{V}$ is the value matrix after projection in attention module and $\boldsymbol{A}$ is the attention matrix, according to linear algebra, we have,

$$\operatorname{rank}(\boldsymbol{X}^{l+1}) < \min\{\operatorname{rank}(\boldsymbol{V}), \operatorname{rank}(\boldsymbol{A})\}.$$

Thus, we have that $\boldsymbol{X}^{l+1}$ is also a low-rank matrix.

