# OpenReview forum: "Taming Transformer Without Using Learning Rate Warmup"
_ICLR.cc/2025/Conference — ICLR 2025 Poster_

### Official Review · Reviewer_1Y1q · 2024-10-25

**Soundness:** 2
**Presentation:** 3
**Contribution:** 3
**Rating:** 6
**Confidence:** 3

**Summary:**

This paper visualizes the training dynamics of transformer models in both successful and failing cases. Based on these visualizations, the authors observe that the rapid growth of the maximal singular value of the product of the query and key matrices,  W_q^T W_k , is a critical phenomenon associated with model crashes. They provide insights suggesting that when  W_q^T W_k  becomes concentrated along the directions of the top singular vectors, the attention probability matrix becomes sparse and low-rank. They also show crashed models typically exhibit significant spectral energy concentration. Finally, the authors propose a variant of AdamW that addresses this issue by clipping the learning rate to prevent the spectral norm of all parameters from increasing too quickly.

**Strengths:**

This paper provides insightful experiments demonstrating that the rapid growth of the spectral norm of the query-key product matrix (and other parameter matrices) is a crucial factor in model crashes. The study shows that in crashed models, the spectrum of the query-key product matrices often concentrates on the top eigenvalues. This provides novel insight on understanding transformer optimization.

**Weaknesses:**

- The theory part does not enhance the paper as effectively as it could.
    - In Theorem 1, the statement is somewhat vague. Terms like “malignant collapse” and “spectral energy concentration” are not formally defined, making the theorem unclear. Additionally, the proof of Theorem 1, which appears in Appendix D, lacks clarity and coherence. For example, the theorem aims to establish a sufficient condition for malignant collapse, yet the proof assumes the low rank of  W  due to malignant collapse. Furthermore, the proof relies heavily on the assumption of Gaussian distribution.
    - I would personally suggest rewriting Theorem 1 as a highlighted insight or intuition, supported by experimental results, such as Figure 1 and 4.
- In Figure 5, could you elaborate on how the concentrated and low-rank structure of  $\frac{\partial \text{vec}(P)}{\partial \text{vec}(W_q^T W_k)}$  leads  W_q^T W_k  to become low rank? It still requires reshaping the matrix, multiplying the activation gradient, and summing over data from  $\frac{\partial \text{vec}(P)}{\partial \text{vec}(W_q^T W_k)}$  to the update of  W_q^T W_k, so this implication is not straightforward.
- Could you clarify how one would generally set the parameter $\tau$ for an unseen model, and how $\tau$ might vary as the model scales up?
- Additionally, does benign collapse occur in real training tasks?

**Questions:**

See weakness above

---

> ### Author Response · Authors · 2024-11-23
> **Author responses to Reviewer 1Y1q (part 1/3)**
>
> We would like to thank the reviewer for the constructive comments and suggestions. We are happy to know that you recognize our theoretical contributions as **"This provides novel insight on understanding transformer optimization"**. We are encouraged by your nice comments. We find that more and more researchers are focusing on fantastic applications based on the Transformer model, but the mathematical understanding of the research community to the Transformer, especially its training stability and representation ability, is still not quite enough. We are very eager to make more contributions to the understanding of the Transformer. We appreciate your recognition of our work. It means a lot to us. We will insist on this direction.
>
> In the revised paper, we have added more experimental results and revised the paper according to your suggestions. Hope our revisions will further address your concerns.
>
> &nbsp;
>
> __*1. Reviewer's comments on theory part*__
>
> > - *In Theorem 1, the statement is somewhat vague. Terms like “malignant collapse” and “spectral energy concentration” are not formally defined, making the theorem unclear. Additionally, the proof of Theorem 1, which appears in Appendix D, lacks clarity and coherence. For example, the theorem aims to establish a sufficient condition for malignant collapse, yet the proof assumes the low rank of W due to malignant collapse. Furthermore, the proof relies heavily on the assumption of Gaussian distribution.*
> > - *I would personally suggest rewriting Theorem 1 as a highlighted insight or intuition, supported by experimental results, such as Figure 1 and 4.*
>
> __Our response__: thank you for your suggestion. According to your suggestions, we have carefully revised the paper, especially have completed the following revisions.
>
> 1. We have clearly and formally defined the term “malignant collapse” and “spectral energy concentration” in the revised vision of our submission.
>
> 2. We have rewritten Theorem 1 to support the insights and intuition in Figure 1 and Figure 4.
>
> 3. We have clarified the assumptions in Appendix C and Appendix D.
>
>    That is true that our proof is based on some assumption of the matrix form $\boldsymbol{W} = \boldsymbol{W}_q^{\top} \boldsymbol{W}_k$ and the Gaussian distribution of $\boldsymbol{x}$ because without these assumptions, according to our knowledge, it is hard for current mathematical tools to prove these theorems. Meanwhile, after LayerNorm, $\boldsymbol{x}$ can generally be considered as a Gaussian distribution. Our proof is motivated by [1, 2].
>
> We kindely invite the reviewer to check the revisions. We rewrite some parts as suggested by you. Thank you for your comments. They really help us strength the paper.
>
> [1] Vershynin, Roman. *High-dimensional probability: An introduction with applications in data science*. Vol. 47. Cambridge university press, 2018.
>
> [2] Horn, Roger A., and Charles R. Johnson. *Topics in matrix analysis*. Cambridge university press, 1994.
>
> &nbsp;

---

> > ### Author Response · Authors · 2024-11-23
> > **Author responses to Reviewer 1Y1q (part 2/3)**
> >
> > __*2. Reviewer's comments on Figure 5*__
> >
> > > - *In Figure 5, could you elaborate on how the concentrated and low-rank structure of  leads* $\frac{\partial \operatorname{vec}(\boldsymbol{P})}{\partial \operatorname{vec}(\boldsymbol{W}_q^{\top} \boldsymbol{W}_k)}$ *to become low rank? It still requires reshaping the matrix, multiplying the activation gradient, and summing over data from  to the update of* $\frac{\partial \operatorname{vec}(\boldsymbol{P})}{\partial \operatorname{vec}(\boldsymbol{W}_q^{\top} \boldsymbol{W}_k)}$, *so this implication is not straightforward.*
> >
> > __Our response__: thank you very much for your comments. We are sorry for the confusion before. We would clarify the following steps. If $\boldsymbol{X}$ is low-rank, then $\boldsymbol{X}^{\top} \otimes \boldsymbol{X}^{\top}$ is also low-rank according to [3] because
> >
> > $$
> > rank(\boldsymbol{X} \otimes \boldsymbol{X}) = rank(\boldsymbol{X}) \times rank(\boldsymbol{X}).
> > $$
> >
> > According to the gradient computation in Proposition 1 in the paper, we have
> >
> > $$
> > \frac{\partial \operatorname{vec}(\boldsymbol{P})}{\partial \operatorname{vec}(\boldsymbol{W}_q^{\top} \boldsymbol{W}_k)} = \boldsymbol{X}^{\top} \otimes \boldsymbol{X}^{\top},
> > $$
> >
> > thus, we know that $\frac{\partial \operatorname{vec}(\boldsymbol{P})}{\partial \operatorname{vec}(\boldsymbol{W}_q^{\top} \boldsymbol{W}_k)}$ is also low-rank. Meanwhile, it should be noted that the spectral energy of $\frac{\partial \operatorname{vec}(\boldsymbol{P})}{\partial \operatorname{vec}(\boldsymbol{W}_q^{\top} \boldsymbol{W}_k)}$ is over-concertrated---this means that the gradient update will largely change $\boldsymbol{W}_q^{\top} \boldsymbol{W}_k$, thus $\boldsymbol{W}_q^{\top} \boldsymbol{W}_k$ will have high probability to be low-rank. In our paper, we have proved that very low rank and large singular value of $\boldsymbol{W}_q^{\top} \boldsymbol{W}_k$ will lead to the attention map $\boldsymbol{A}$ over-concentrated, and hence becomes to be a sparse yet low-rank matrix. Finally, an over-concertrated $\boldsymbol{A}$ will lead to $\boldsymbol{X}^{l+1}$ to be low-rank.
> >
> > [3] Petersen, Kaare Brandt, and Michael Syskind Pedersen. "The matrix cookbook." *Technical University of Denmark* 7.15 (2008): 510.
> >
> > &nbsp;

---

> > > ### Author Response · Authors · 2024-11-23
> > > **Author responses to Reviewer 1Y1q (part 3/3)**
> > >
> > > __*3. Reviewer's comments on the parameter $\tau$*__
> > >
> > > > - *Could you clarify how one would generally set the parameter $\tau$ for an unseen model, and how  might vary as the model scales up?*
> > >
> > > __Our repsonse__: In our previous submission, we use $\tau = 0.004$ for all Vision Transformer experiments. Generally, we believe that we should use a smaller $\tau$ for a larger models. However, $\tau$ should not be too small.
> > >
> > > &nbsp;
> > >
> > > ***4. Reviewer's comments on benign collapse***
> > >
> > > > - *Additionally, does benign collapse occur in real training tasks?*
> > >
> > > **Our response**: Thank you very much for your question. This is a valuable and interesting question. The answer is yes, it does exist.  However, in our experiments, we only observed this phenomenon in ViT. And it only happens in several heads of the first Transformer block in the ViT model, but we did not observe this phenomenon in GPT.
> > >
> > > &nbsp;
> > >
> > > We sincerely thank the reviewer 1Y1q for your valuable suggestions and interesting questions, which have helped us improve our paper. **Hope our clarifications could address your concerns. We are open to have any additional discussion.**
> > >
> > > &nbsp;

---

> > > > ### Comment · Reviewer_1Y1q · 2024-11-27
> > > >
> > > > Thank you for your reply. I have some additional concerns:
> > > >
> > > > - **Theorem 1 Writing:** The presentation of Theorem 1 is still not formal.
> > > >   - For the first bullet point, I recommend avoiding the use of $\approx$ symbols.
> > > >   - For the second bullet point, please clearly define how large $s$ is.
> > > >   - My concern is that if the theorem is not written formally, it becomes unclear what exactly the proof is trying to establish. Please ensure the theoretical components are rigorously structured if you aim to present them as formal theory.
> > > >
> > > > - **Explanation of Figure 5:** Your explanation of Figure 5 is unclear to me. Could you explain step by step what you mean by: "noted that the spectral energy of [...] is over-concentrated---this means that the gradient update will largely change [...], thus [...] will have high probability to be low-rank"? A more detailed explanation would help clarify this point. Formal explanations like equations are welcome.

---

> > > > > ### Author Response · Authors · 2024-11-28
> > > > > **Author response to Reviewer 1Y1q**
> > > > >
> > > > > Thank you very much for your careful reading. We are working on your suggestions and comments, and we expect to return our responses in one or two days.

---

> ### Author Response · Authors · 2024-11-30
> **Author response to Reviewer 1Y1q (part 1)**
>
> We sincerely thank reviewer 1Y1q for your careful reading and your constructive suggestions. We take your suggestions seriously. We present our responses in the following. We have rewritten Theorem 1 and updated its associated section in Appendix.  We will also add a section in appendix to interprete Figure 5 in detail in the revised version. Since that at this moment we cannot update the submission, we will update our submission after the OpenReview system allows us to do it.
>
> For the first question, we have rewritten Theorem 1 in a more formal way. For the second question, we have given a detailed explanation for each arrow in Figure 5.
>
> We hope that you are satisfied with our responses.  Thank you again for your comments.
>
> &nbsp;
>
> **Theorem (Malignant Entropy Collapse)**
> Let $ \boldsymbol{P} = \boldsymbol{X}^{\top} \boldsymbol{W} \boldsymbol{X}$ and $\boldsymbol{A} = \operatorname{softmax}(\frac{\boldsymbol{P}}{\sqrt{d_q}})$, where ${\boldsymbol{W}} ={\boldsymbol{W}_q}^{\top} {\boldsymbol{W}_k} \in \mathcal{R}^{d\times d}$.  Suppose that the following two conditions are simultaneously satisified:  a) $ \boldsymbol{X}$ is a low-rank matrix; b) $\boldsymbol{W}$ is a low-rank matrix with only a few dominant singular values (e.g., the singular values are greater than $ C_0 \cdot \sqrt{d_q}$ where $C_0 \gg 1$ is a constant).
> Then, $\boldsymbol{A}$ will have exhibit the following properties in high probability:
> 1) **Sparsity:** $\boldsymbol{A}$ becomes sparse, i.e., the number of non-zero elements in each column of $ \boldsymbol{A}$ is small;
> 2) **Low-rank:** $\boldsymbol{A}$ becomes approximately low-rank.
>
>
>
> &nbsp;

---

> ### Author Response · Authors · 2024-11-30
> **Author response to Reviewer 1Y1q (part 2)**
>
> #### **Illustration of Figure 5**
>
> We illustrate Figure 5 as follows. We explain each arrow in Figure 5.
>
> ##### **1) Illustration of Arrow 1:**
>
> According to the property of Kronecker Product, we have
>
> $$
> rank(\boldsymbol{X} \otimes \boldsymbol{X}) = rank(\boldsymbol{X}) \times rank(\boldsymbol{X}).
>
> Since $\boldsymbol{X}$ is low-rank, then $\boldsymbol{X} \otimes \boldsymbol{X}$ is also low-rank. In the following, we will also prove the singular values of $\boldsymbol{X} \otimes \boldsymbol{X}$ will also strengthen the concertration of spectral energy into some directions with large singular values.
>
> &nbsp;
>
> ##### **2) Illustration of Arrow 2:**
>
> According to the computation of Jacobian matrix, we have
>
> $$
> \frac{\partial \operatorname{vec}(\boldsymbol{P})}{\partial \operatorname{vec}({\boldsymbol{W}_q}^{\top}{\boldsymbol{W}_k} )} =  \boldsymbol{X}^{\top} \otimes \boldsymbol{X}^{\top},
> $$
>
> where $\boldsymbol{P} = \boldsymbol{X}^{\top} \boldsymbol{W}_q^{\top} \boldsymbol{W}_k \boldsymbol{X}^{\top}.$
>
> &nbsp;
>
> ##### **3) Illustration of Arrow 3:**
>
> Let $\boldsymbol{X} \in \mathbb{R}^{m \times n}$ be a matrix with rank $r \leq \min(m,n)$. Denote the singular values of $\boldsymbol{X}$ as $\sigma_1 \geq \sigma_2 \geq \cdots \geq \sigma_r > 0$ and $\sigma_{r+1} = \cdots = \sigma_{\min(m,n)} = 0$.
>
> **Definition (Singular Values of Kronecker Product)**
> For a matrix $\boldsymbol{X}$, define $\Lambda(\boldsymbol{X})$ as the set of all possible products of its singular values, i.e.,$ \Lambda(\boldsymbol{X}) = \\{\sigma_i \sigma_2 : 1\leq i,j \leq r\\} $
>
> **Theorem (Singular Values of Kronecker Product)**
> For a low-rank matrix $\boldsymbol{X} \in \mathbb{R}^{m \times n}$ with rank $r$, the singular values of $\boldsymbol{X} \otimes \boldsymbol{X}$ are precisely the elements in $\Lambda(\boldsymbol{X})$. More formally, let $\{\mu_k\}$ be the singular values of $\boldsymbol{X} \otimes \boldsymbol{X}$. Then$ \{\mu_k\} = \{\sigma_i \sigma_j \quad \text{for some } 1 \leq i,j \leq r\}.$
>
> **Proof**. Consider the singular value decomposition (SVD) of $\boldsymbol{X} = \boldsymbol{U}\boldsymbol{\Sigma} \boldsymbol{V}^T$, where:
>
>  $\boldsymbol{U} \in \mathbb{R}^{m \times m}$ is an orthogonal matrix, $\boldsymbol{V} \in \mathbb{R}^{n \times n}$ is an orthogonal matrix, and $\boldsymbol{\Sigma} = \text{diag}(\sigma_1, \ldots, \sigma_r, 0, \ldots, 0)$.
>
> The Kronecker product $\boldsymbol{X} \otimes \boldsymbol{X}$ can be expanded as:
> $\boldsymbol{X} \otimes \boldsymbol{X} = (\boldsymbol{U}\boldsymbol{\Sigma} \boldsymbol{V}^T) \otimes (\boldsymbol{U}\boldsymbol{\Sigma} \boldsymbol{V}^T) = (\boldsymbol{U} \otimes \boldsymbol{U})(\boldsymbol{\Sigma} \otimes \boldsymbol{\Sigma})(\boldsymbol{V}^T \otimes \boldsymbol{V}^T)$
>
> The singular values of $\boldsymbol{U} \otimes \boldsymbol{U}$ and $\boldsymbol{V}^T \otimes \boldsymbol{V}^T$ are all 1, as they are composed of orthogonal matrices.
>
> Therefore, the singular values of $\boldsymbol{X} \otimes \boldsymbol{X}$ are precisely the elements of $\boldsymbol{\Sigma} \otimes \boldsymbol{\Sigma}$, which are exactly $\{\sigma_i \sigma_j : 1 \leq i,j \leq r\}$.
>
> In summary, the Singular Values of Kronecker Product $\boldsymbol{X} \otimes \boldsymbol{X}$ will also strengthen the concertration of spectral energy into some directions with large singular values. In this way, the singular values of  $\frac{\partial \operatorname{vec}(\boldsymbol{P})}{\partial \operatorname{vec}({\boldsymbol{W}_q}^{\top}{\boldsymbol{W}_k} )}$ will also concentrate to a few directions.
>
> When we update $\boldsymbol{W}_q^{\top} \boldsymbol{W}_k$ according to the following equation,
>
> $$
> {\operatorname{vec}(\boldsymbol{W}_q^{\top} \boldsymbol{W}_k)}_n =  {\operatorname{vec}(\boldsymbol{W}_q^\top \boldsymbol{W}_k)}_o - \alpha \frac{\partial \mathcal{L}}{\partial \operatorname{vec}(\boldsymbol{P})} \frac{\partial \operatorname{vec}(\boldsymbol{P})}{\partial \operatorname{vec}({\boldsymbol{W}_q}^{\top}{\boldsymbol{W}_k})_o}
> $$
>
> where $\mathcal{L}$ is the loss function, $\alpha$ is the step size, and ${\operatorname{vec}(\boldsymbol{W}_q^{\top} \boldsymbol{W}_k)}_n$ is the newly updated ${\operatorname{vec}(\boldsymbol{W}_q^{\top} \boldsymbol{W}_k)}$. Since the singular values of $\frac{\partial \operatorname{vec}(\boldsymbol{P})}{\partial \operatorname{vec}({\boldsymbol{W}_q}^{\top}{\boldsymbol{W}_k} )}$ will also concentrate to a few directions, the update will lead to the singular values of $\boldsymbol{W}_q^{\top} \boldsymbol{W}_k$ tends to concentrate.
>
> &nbsp;

---

> ### Author Response · Authors · 2024-11-30
> **Author response to Reviewer 1Y1q (part 3)**
>
> ##### **Illustration of Arrow 4**
>
> Our Theorem 1 in the paper is to demonstrate that when $\boldsymbol{X}$ is low-rank and $\boldsymbol{W}_q^{\top} \boldsymbol{W}_k$ a low-rank matrix with only a few dominant singular values, the attention map $\boldsymbol{A}$ will have a large probability to be a sparse yet low-rank matrix.
>
> &nbsp;
>
> ##### **Illustration of Arrow 5**
>
> Since $\boldsymbol{X}^{l+1} = \boldsymbol{V} \boldsymbol{A} $, where $\boldsymbol{A}$ is the attention matrix, and $\boldsymbol{V}$ is the values after projection in attention module, according to linear algebra, we have,
>
> $$
> \text{rank}(\boldsymbol{X}^{l+1}) \leq \min( \text{rank}(\boldsymbol{V}), \text{rank}(\boldsymbol{A})).
> $$
>
> Since that $\boldsymbol{A}$ is a sparse yet low-rank matrix, then we have that $\boldsymbol{X}^{l+1}$ is also a low-rank matrix.

---

### Official Review · Reviewer_9a7P · 2024-10-29

**Soundness:** 3
**Presentation:** 3
**Contribution:** 3
**Rating:** 6
**Confidence:** 3

**Summary:**

The paper presents a method to eliminate learning rate warmup for training transformes. It begins by analyzing training dynamics, specifically observing the norm of certain parameters to understand behavior patterns when training is successful versus when it crashes. The authors note that, in the presence of warmup, the parameter norms tend to increase steadily, while they diverge in failed training runs. However, this analysis does not introduce new insights, as it’s known that parameter norms typically blow up when training crashes.
The authors argue that warmup can be bypassed by controlling the spectral energy concentration (SEC), which maintains the singular values of the product $W_q^TW_k$​ thus allowing successful training. To achieve this, they introduce a modified version of the AdamW optimizer using a power iteration-based step to estimate the largest singular values of the $W_q^TW_k$ matrix. Experiments with the proposed method show that it can perform similar to warmup.

**Strengths:**

1 .The authors propose an alternate method for transformer training that does not need learning rate warmup.

**Weaknesses:**

1. The choice of parameter norms for tracking training dynamics seems arbitrary. It’s already established in literature that parameter norms diverge when training fails, so this observation does not seem novel [1]. The paper could benefit from a deeper analysis or rationale for the specific parameters chosen, or alternatively, from an exploration of novel insights that could provide a more compelling argument.
2. While the authors claim that the attention maps are sparse and low-rank, the plots are difficult to interpret. Hence it is impossible to make conclusions based on the provided plots. It might help to change the scaling of the plots to help visualization or simply estimate the rank of the attention maps.
3. My major concern is that in the proposed algorithm that controls the step size to ensure the SEC does not collapse, only ends up modifying the step size $\alpha$. It seems to me that this is another way of performing learning rate warmup and the analysis further justifies the use of warmup as it may help prevent SEC from collapsing. In order to disentangle these ideas, the authors might need to discuss how the modified $\alpha_t$ looks over time (does it resemble warmup, which seems to be the case).
Experiments also suggest that the proposed method is in fact very similar to warmup and might be a different parameterization offering the same benefit as warmup.
4. Further, while replacing warmup the authors introduce an additional power iteration step which is computationally more expensive and adds to each gradient update step, while learning rate warmup does not require this additional step.


[1] Pennington, Jeffrey, Samuel Schoenholz, and Surya Ganguli. "Resurrecting the sigmoid in deep learning through dynamical isometry: theory and practice." Advances in neural information processing systems 30 (2017).

[2] Defazio, Aaron, et al. "When, why and how much? adaptive learning rate scheduling by refinement." arXiv preprint arXiv:2310.07831 (2023).

**Questions:**

See above.

---

> ### Author Response · Authors · 2024-11-23
> **Author responses to Reviewer 9a7P (part 1/3)**
>
> We would like to thank the reviewer for the constructive suggestions. To address your concerns, we have added more experiments, clarified our key contributions, and revised the paper according to your suggestions. We really hope you will be satisfied with our revisions.
>
> &nbsp;
>
> *__Reviewer's comments on paper's strengths__*
>
> > *1 .The authors propose an alternate method for transformer training that does not need learning rate warmup.*
>
> __Our response__: Thank you very much for your comments. We would like to clarify our contributions in this paper as follows.
>
> 1. We visualize the training dynamics of Transformers that train successfully or unsuccessfully and summarize two important observations from unsuccessful training: a) the rank of the attention map matrix tends to be very small and the entropy of the attention probability matrix tends to 0; and b) $\sigma_1({\boldsymbol{W}_q}^{\top} \boldsymbol{W}_k)$ increases rapidly to very large value.
>
> 2. We present rigid theoretical analysis for phenomina that are frequently happened during the Transformer training, finding that the Jacobian matrix $\frac{\partial \operatorname{vec}(\boldsymbol{P})}{\partial \operatorname{vec}({\boldsymbol{W}_q}^{\top}{\boldsymbol{W}_k} )} =  \boldsymbol{X}^{\top} \otimes \boldsymbol{X}^{\top}$, which implies that the gradient of ${\boldsymbol{W}_q}^{\top}{\boldsymbol{W}_k}$ is largely dominated by the rank of $\boldsymbol{X}^{\top} \otimes \boldsymbol{X}^{\top}$.
>
> 3. We reveal that the key problem in the model crash is the spectral energy concentration (SEC) of $\ {\boldsymbol{W}_q}^{\top} \boldsymbol{W}_k$}, which will cause the attention map to be a sparse yet low-rank matrix, which corresponds to a malignant entropy collapse.
>
> 4. Motivated by Weyl's inequality, We introduce a novel strategy to address the problem of spectral energy concentration of ${\boldsymbol{W}_q}^{\top} \boldsymbol{W}_k$ by controlling the rapid growth of singular values, and verify that our strategy leads to a stable training process.
>
> In summary, our main contributions are to provide a deep understanding of the training dynamics of Transformer and to reveal the underlying reasons for the model crash with rigid theoritical analyse. The fourth contribution well justfies our observation in model crash when malignant collapse happens.
>
> In the revised version of our submission, we have added more experiments to support our theoretical contributions. Hope our revisions will address your concerns well.
>
> &nbsp;

---

> > ### Author Response · Authors · 2024-11-23
> > **Author responses to Reviewer 9a7P (part 2/3)**
> >
> > *__1. Reviewer's comments on tracking training dynamics__*
> >
> > > *The choice of parameter norms for tracking training dynamics seems arbitrary. It’s already established in literature that parameter norms diverge when training fails, so this observation does not seem novel [1]. The paper could benefit from a deeper analysis or rationale for the specific parameters chosen, or alternatively, from an exploration of novel insights that could provide a more compelling argument.*
> > >
> > > *[1] Pennington, Jeffrey, Samuel Schoenholz, and Surya Ganguli. "Resurrecting the sigmoid in deep learning through dynamical isometry: theory and practice." Advances in neural information processing systems 30 (2017).*
> > >
> > > *[2] Defazio, Aaron, et al. "When, why and how much? adaptive learning rate scheduling by refinement." arXiv preprint arXiv:2310.07831 (2023).*
> >
> > __Our response__: Thanks for your comments and recommended literatures. In [1], Jeffrey et al. discussed a concept, called dynamic isometry.  It mentioned that the "dynamical isometry" is defined as all singular values of the Jacobian concentrate near 1. It means for a linear layer, i.e., $\boldsymbol{y} = \boldsymbol{W} \boldsymbol{x}$, its Lipschitz constant is enforced to 1 under $L_2$ norm.  In this work, we do not enforce such a strong condition to the weight matrix. Instead, we only require that the singular value of $\boldsymbol{W}$ does not increase too fast in each update. We believe that the dynamic isometry is an over-strict condition, it may limit the capacity of the network.  For example, for a single-head self-attention, the forward process is defined as
> > $$
> > \boldsymbol{Y} = \boldsymbol{W}_v \boldsymbol{X} \boldsymbol{A},
> > $$
> > where $\boldsymbol{P} = \boldsymbol{X}^{\top} {\boldsymbol{W}_q}^{\top} {\boldsymbol{W}_k} {\boldsymbol{X}},  \quad \boldsymbol{A} = \operatorname{softmax}(\frac{\boldsymbol{P}}{\sqrt{d_q}}).$
> >
> > Its Jacobian matrix is computed as
> >
> > $$
> > \frac{\partial \text{vec}(\boldsymbol{Y})}{\partial \text{vec}(\boldsymbol{X})} = (\boldsymbol{A}^\top \otimes \boldsymbol{W}_v) + (\boldsymbol{I}_n \otimes \boldsymbol{W}_v\boldsymbol{X}) \left( \frac{ \text{diag}(\text{vec}(\boldsymbol{A})) - (\boldsymbol{A} \otimes \boldsymbol{I}_n) \text{diag}(\text{vec}(\boldsymbol{A}))}{\sqrt{d_q}} \right) \left(
> >   (\boldsymbol{X}^{\top}{\boldsymbol{W}_q}^{\top}{\boldsymbol{W}_k} \otimes \boldsymbol{I}_n)\boldsymbol{K} + (\boldsymbol{I}_n \otimes \boldsymbol{X}^{\top}{\boldsymbol{W}_q}^{\top}{\boldsymbol{W}_k})
> >  \right).
> > $$
> >
> > The Jacobian matrix of $\frac{\partial \text{vec}(\boldsymbol{Y})}{\partial \text{vec}(\boldsymbol{X})}$ is very complex. It is hard to simply enforce the dynamical isometry.  Our method does not assume any form of the Jacobian matrix $\frac{\partial \text{vec}(\boldsymbol{Y})}{\partial \text{vec}(\boldsymbol{X})}$.
> >
> > We have read many works from Aaron Defazio and appreciate very much. However, their work starts from numerical or convex optimization to design the learning rate schedule; whereas our work is from matrix calculus and high-dimensional statistics.
> >
> > &nbsp;
> >
> > ***2. Reviewer's comments on plots in the paper***
> >
> > > *While the authors claim that the attention maps are sparse and low-rank, the plots are difficult to interpret. Hence it is impossible to make conclusions based on the provided plots. It might help to change the scaling of the plots to help visualization or simply estimate the rank of the attention maps.*
> >
> > **Our response**: Thank you very much for your suggestions. To help the readers understand the figures better, we add color bar at the right side of Figure 3. Figure 2 in our submission is actually a flash that consists tens of frames. It illustrates the changing process of attention maps. When we mention "the attention maps are sparse and low-rank", as shown in the middle of Figure 2 that, only a few vertical lines associate large probability but the rest lines are almost vanishing. Thank you very much for your suggestions. We kindly invite the reviewer to check the newly updated **Figure 3**  in the revised paper.
> >
> > &nbsp;

---

> > > ### Author Response · Authors · 2024-11-23
> > > **Author responses to Reviewer 9a7P (part 3/3)**
> > >
> > > ***3. Reviewer's comments on the step size to ensure the Spectral Energy Concentration (SEC)***
> > >
> > > > *My major concern is that in the proposed algorithm that controls the step size to ensure the SEC does not collapse, only ends up modifying the step size . It seems to me that this is another way of performing learning rate warmup and the analysis further justifies the use of warmup as it may help prevent SEC from collapsing. In order to disentangle these ideas, the authors might need to discuss how the modified  looks over time (does it resemble warmup, which seems to be the case). Experiments also suggest that the proposed method is in fact very similar to warmup and might be a different parameterization offering the same benefit as warmup.*
> > >
> > > **Our response**: Thank you very much for your comments. Learning rate strategy or schedule [3] is very important for deep learning training. Actually in our method, **an important property of our method is that in different layer, our strategy will adaptively choose different learning rate**. According to your suggestion, in **Figure 14** of the **Appendix M** in the revised paper, we have shown how the actual step size varies over time. Please refer to the revised version of our submission. We kindly invite the reviewer to check in the newly added **Figures 14** in the revised paper. Although the curve of the acutal step size looks like a learning rate warmup curve, we observe that different layers have different learning rate curves. For instance, we observed that the shallower layers are more likely leading to a larger update of weight matrix and typically require a smaller learning rate. However, in the standard training settings, the learning rates for all layers are set the same.
> > >
> > > We really hope you will be satified with our revision. We are open to have any additional discussion.
> > >
> > > [3] Defazio, A., Yang, X. A., Mehta, H., Mishchenko, K., Khaled, A., & Cutkosky, A. (2024). The road less scheduled. *arXiv preprint arXiv:2405.15682*.
> > >
> > > &nbsp;
> > >
> > > ***4. Reviewer's comments on computational complexity***
> > >
> > > > *Further, while replacing warmup the authors introduce an additional power iteration step which is computationally more expensive and adds to each gradient update step, while learning rate warmup does not require this additional step.*
> > >
> > > **Our response**: Thank you very much for your comments. The power interation step is very fast in each update step. In power iteration, we only need two times of matrix-vector products instead of a matrix-matrix product. For a matrix $A\in \mathcal{R}^{n\times n}$, the computational complexity of power iteration is $\mathcal{O}(n^2)$ instead of $\mathcal{O}(n^3)$ in SVD. Learning rate warmup is a very good and effective method, but we believe that we should not stay at the origin and deny any new develpment with potential significance. Hope you will also recognize that our mathemtical study is meaningful.
> > >
> > > We sincerely thank the reviewer 9a7P for your valuable suggestions, which have helped us strengthen our paper. **Hope our clarifications and the new experiments could address your concerns well. We are open to have any additional discussions.**

---

> > > > ### Comment · Reviewer_9a7P · 2024-11-24
> > > > **Reponse to authors**
> > > >
> > > > I thank the authors for taking the time and providing clarifications for all my questions.
> > > >
> > > > 1. While I agree that dynamical isometry is a stronger condition, I believe the a discussion of such related work in the context of identifying various parameters to track in the paper will make the paper stronger and I would urge the authors to include such a discussion.
> > > > 2. Thank you for the clarification on the figure.
> > > > 3. The provided Figure 14 addresses my concern about similarity to warmup.
> > > > 4. The computational cost of a single power iteration step is not an issue.
> > > >
> > > > I believe most of my concerns have been address and I will update my score accordingly.

---

> > > > > ### Author Response · Authors · 2024-11-25
> > > > > **Author response to Reviewer 9a7P**
> > > > >
> > > > > Thank you for your positive feedbacks. We genuinely appreciate your insightful and constructive comments that greatly enhanced the quality of our work.

---

### Official Review · Reviewer_YrRp · 2024-10-30

**Soundness:** 3
**Presentation:** 2
**Contribution:** 2
**Rating:** 6
**Confidence:** 3

**Summary:**

The paper analyzes a training instability associated with the self-attention layer of a Transformer by studying its training dynamics. Specifically, the paper identifies a ```Spectral Energy Concentration``` condition where-in the weights of the product of query transpose key matrix shows a rank collapse. The paper proposes to bound the base learning rate per step when the optimizer (AdamW specifically) proposes very large update vectors. The paper shows how the need to use learning rate warmup can be  relaxed for training Transformers. Empirical results are provided to support analysis and proposed algorithm.

**Strengths:**

- The analysis of the  query (transpose), key weight matrix product  is interesting. Specifically, using the singular values of this product matrix to show collapse is interesting
- Application of Weyl's inequality to the update equation for the optimizer to stabilize Transformer training is a nice development
- The biggest contribution of this paper is that the proposed algorithm built on analysis frees up a practitioner to use  learning rate warmup.

**Weaknesses:**

- The paper contains limited empirical results. Specifically, the experimental setup used to study the algorithm is small and no results are provided with larger models (0.5B, 1B, 3B). This limits the contribution as its not clear how the proposed algorithm will work with larger models (albeit can still be considered ``small'') where training instabilities are more apparent

- Related to above, the QK layer norm paper by Deghani et al. (2023) show that training instabilities occur at 8B for vision transformers (ViTs). So, I wonder whether the new update along with no learning rate warmup will work as advertised

- The paper proposes ``Spectral Energy Concentration (SEC) '' to quantify collapse. Wouldn't a smooth rank measure also provide the similar  information while being easier to interpret? Furthermore, this contribution is not novel as large attention logits being a problem has been observed in literature below by Deghani et al (referred to in the paper) and by [Worstman et al.](https://arxiv.org/abs/2309.14322).

- The argument in Section 3.2 is not very clear. Especially the reader could use some help with understanding why the rank of X.T being small would imply the rank of X.T \otimes X.T is still small. The equation shows a square relationship. Firming up the arguments here would be very helpful to the reader

**Questions:**

- Would the proposed optimizer update work at reasonable scales like the ones considered by [Worstman et al.](https://arxiv.org/abs/2309.14322)?

- How different is the instability shown by analyzing weights of Q.T, K product different from logit norm growth shown previously in literature?

- (clarification request) The paper describes the product matrix in line 345 in general as a non-symmetric positive quasi-definite
square matrix. A reference to what these technical terms mean and/or definition in the appendix would help clarify what this means to the reader. The reviewer was unable to come up with a definition of quasi-definite for non-symmetric matrices but this maybe perhaps because I missed looking at an appropriate reference.

The concerns raised in the review is that the work has limited novelty and also limited empirical support. Therefore, the initial decision for this paper is a reject.

---

> ### Author Response · Authors · 2024-11-23
> **Author responses to Reviewer YrRp (part 1/4)**
>
> We would like to express our thanks to the reviewer for the constructive suggestions and are happy to know you recognize our theoretical contributions. We are encouraged by your comments. We will continue to use rigorous mathematical analysis to solve practical deep learning challenges. We believe that the theoretical insights not only explain the existing phenomena but also can bring in improvements in training methodology.
>
> To address your concerns, we have added experiments on nanoGPT-large (0.77B) and ViT-g (1B parameters), clarified some confusing sentences, and revised the paper carefully according to your suggestions. Hope you will be satisfied with our revisions.
>
> &nbsp;
>
> *__1.1 Reviewer's comments on large models (0.5B, 1B, 3B)__*
>
> - > *The paper contains limited empirical results. Specifically, the experimental setup used to study the algorithm is small and no results are provided with larger models (0.5B, 1B, 3B). This limits the contribution as its not clear how the proposed algorithm will work with larger models (albeit can still be considered ``small'') where training instabilities are more apparent*
>
> __Our response__: Thank you very much for your comments. According to your suggestions, we have tested our $\text{AdamW}^2$ on nanoGPT-large (0.77B) and ViT-g (1B parameters). We found that our $\text{AdamW}^2$ can stably train the model without learning rate warmup. We have added these experiments into the revised version of the paper. We kindly invite you to check the newly added **Figures 11, 12, and 13** in the **Appendix J, K** in the revised version of our paper.
>
> &nbsp;
>
> *__1.2 Reviewer's comments on QK layer norm__*
>
> - > *Related to above, the QK layer norm paper by Deghani et al. (2023) show that training instabilities occur at 8B for vision transformers (ViTs). So, I wonder whether the new update along with no learning rate warmup will work as advertised*
>
> __Our response__: Thank you for your suggestion. _Our analysis is generalizable to any Transformer network._ Our analysis does not assume the model uses QK norm. We have trained much larger models with our stablizing strategy $\text{AdamW}^2$, and observed that the model smoothly converges without using learning rate warmup. We believe that the newly added experiments can further verify our theoretical analysis. Hope you will be satisfied with our reply.
>
> &nbsp;

---

> > ### Author Response · Authors · 2024-11-23
> > **Author responses to Reviewer YrRp (part 2/4)**
> >
> > *__1.3 Reviewer's comments on Spectral Energy Concentration (SEC)__*
> >
> > - > *The paper proposes ``Spectral Energy Concentration (SEC) '' to quantify collapse. Wouldn't a smooth rank measure also provide the similar information while being easier to interpret? Furthermore, this contribution is not novel as large attention logits being a problem has been observed in literature below by Deghani et al (referred to in the paper) and by [Worstman et al.](https://arxiv.org/abs/2309.14322).*
> >
> > __Our response__:
> >
> > According to [1], the logit is defined as $z_{ij}=\boldsymbol{q}_i^{\top} \boldsymbol{k}_i/\sqrt{d_h}$. We read the paper (Wortsman et al. 2023), which describes the attention logit growth as "*Dehghani et al.  observed that the attention logits z became large, which they refered to as attention logit growth.*". Briefly, __the max value of the attention logit that increases along with the training steps is called attention logit growth, however, we argue that the largest attention logit increases cannot reflect that all spetral energy concentrates to a few directions__.  They plot the max attention logit along with training steps. Practical observations and findings are very valuable and important. However, our work devotes to mathematically analyze why the spectral energy of $\boldsymbol{W}_q^{\top} \boldsymbol{W}_k$ will concentrate to a few directions and why the spectral energy concentration will lead to model crash.
> >
> > **Our paper, by a rigid theoretical analysis, reveals that the Spectral Energy Concentration (SEC) of $\boldsymbol{W}_q^{\top} \boldsymbol{W}_k$ makes the attention map matrix to be sparse yet low-rank and it is the inner reason leading to model crash.** This is totally different from [1]. We believe that both practical and theoretical research are valuable and important. Our findings with rigid theoretical justification are of meaningful contributions and worthwhile to publish.
> >
> > In addition, the mentioned "smooth rank measure" sounds very interesting and is appealing to be investivated as future work. Thanks the reviewer for the valuable suggestion.
> >
> > [1] Wortsman, Mitchell, et al. "Small-scale proxies for large-scale transformer training instabilities." *arXiv preprint arXiv:2309.14322* (2023).
> >
> > &nbsp;
> >
> > **1.4 *Reviewer's comments on* $rank(\boldsymbol{X}^{\top} \otimes \boldsymbol{X}^{\top})$**
> >
> > - > *The argument in Section 3.2 is not very clear. Especially the reader could use some help with understanding why the rank of X.T being small would imply the rank of X.T \otimes X.T is still small. The equation shows a square relationship. Firming up the arguments here would be very helpful to the reader*
> >
> > __Our response__: Thanks for your comments. According to p.60 in [3],
> >
> > for a matrix $\boldsymbol{X} \in {\mathcal{R}}^{m\times n}$, we have
> >
> > $$
> > rank(\boldsymbol{X} \otimes \boldsymbol{X}) = rank(\boldsymbol{X}) \times rank(\boldsymbol{X})
> > $$
> >
> > We have revised this part according to your comments. We appreciate your suggestion.
> >
> > [3] Petersen, Kaare Brandt, and Michael Syskind Pedersen. "The matrix cookbook." *Technical University of Denmark* 7.15 (2008): 510.
> >
> > &nbsp;

---

> > > ### Author Response · Authors · 2024-11-23
> > > **Author responses to Reviewer YrRp (part 3/4)**
> > >
> > > *__2.1 Reviewer's question 1__*
> > >
> > > - > *Would the proposed optimizer update work at reasonable scales like the ones considered by [Worstman et al.](https://arxiv.org/abs/2309.14322)?*
> > >
> > > __Our response__: Thank you for your question. We have tested our $\text{AdamW}^2$ on nanoGPT-large (0.77B) and ViT-g (1B). The experiments confirmed that our $\text{AdamW}^2$ can stably train the large models without learning rate warmup. We kindly invite the reviewer to check the newly added Figures 11, 12, and 13 in the Appendix J, K in the revised paper.
> > >
> > > &nbsp;
> > >
> > > *__2.2 Reviewer's question 2__*
> > >
> > > - > *How different is the instability shown by analyzing weights of Q.T, K product different from logit norm growth shown previously in literature?*
> > >
> > > **Our response**:  With rigid mathematical derivation, We revealed that the Spectral Energy Concentration (SEC) of $\boldsymbol{W}_q^{\top} \boldsymbol{W}_k$ makes the attention map matrix to be sparse yet low-rank and thus is the inner reason leading to model crash. Our theoretical findings well justified the weird phenomena observed on the attention map matrix.
> > > On contrary, as replied in 1.3 that, the logit norm growth does not reveal the hidden reason and mathematical mechanism behind the model crash. Thus, our work is still meaningful and worthwhile to publish.
> > >
> > > &nbsp;

---

> > > > ### Author Response · Authors · 2024-11-23
> > > > **Author responses to Reviewer YrRp (part 4/4)**
> > > >
> > > > ***2.3 Reviewer's question 3***
> > > >
> > > > - > (*clarification request) The paper describes the product matrix in line 345 in general as a non-symmetric positive quasi-definite square matrix. A reference to what these technical terms mean and/or definition in the appendix would help clarify what this means to the reader. The reviewer was unable to come up with a definition of quasi-definite for non-symmetric matrices but this maybe perhaps because I missed looking at an appropriate reference.*
> > > >
> > > > __Our response__: Thank you very much for your professionalism and rigorousness. When we mention $\boldsymbol{W}_q^{\top} \boldsymbol{W}_k$ is  a non-symmetric positive quasi-definite square matrix, we mean it has the following three properties.
> > > >
> > > > 1. $\boldsymbol{W}_q^{\top} \boldsymbol{W}_k$ is not symmetric because generally,  $\boldsymbol{W}_q^{\top} \boldsymbol{W}_k \neq {\boldsymbol{W}_k^{\top} \boldsymbol{W}_q}$
> > > >
> > > > 2. $\boldsymbol{W}_q^{\top} \boldsymbol{W}_k$ is a square matrix and most of its eigenvalues are larger than 0, and only very few are less than 0. That is why it is called positive quasi-definite.
> > > >
> > > > 3. Assume that $\boldsymbol{W}=\boldsymbol{W}_q^{\top} \boldsymbol{W}_k$ is positive definite matrix. If the entries of $\boldsymbol{x}$ are independently and identically sampled from a Gaussian distribution, we can prove that $\mathbb{E}\left[{\boldsymbol{x}_i}^{\top} \boldsymbol{W} \boldsymbol{x}_i\right] \gg \mathbb{E}\left[{\boldsymbol{x}_i}^{\top} \boldsymbol{W} \boldsymbol{x}_j\right]$ when $i\neq j$, see Appendix C for the proof.
> > > >
> > > > The three points have been clearly added in the revised paper. Thank you for your question.
> > > >
> > > > &nbsp;
> > > >
> > > > We sincerely thank reviewer YrRp for the valuable comments and suggestions, which have greatly strengthened our paper.
> > > >
> > > > **Hope our clarifications and the new experiments could address your concerns well. We are open to any additional discussion.**

---

> > > ### Comment · Reviewer_YrRp · 2024-11-25
> > > **Updated draft is good**
> > >
> > > > > 1.3 Reviewer's comments on Spectral Energy Concentration (SEC)
> > >
> > > > >The paper proposes ``Spectral Energy Concentration (SEC) '' to quantify collapse. Wouldn't a smooth rank measure also provide the similar information while being easier to interpret? Furthermore, this contribution is not novel as large attention logits being a problem has been observed in literature below by Deghani et al (referred to in the paper) and by Worstman et al..
> > >
> > > > Our response:
> > >
> > > > According to [1], the logit is defined as $z_{ij}=\boldsymbol{q}_i^{\top} \boldsymbol{k}_i/\sqrt{d_h}$. We read the paper (Wortsman et al. 2023), which describes the attention logit growth as "Dehghani et al. observed that the attention logits z became large, which they refered to as attention logit growth.". Briefly, the max value of the attention logit that increases along with the training steps is called attention logit growth, however, we argue that the largest attention logit increases cannot reflect that all spetral energy concentrates to a few directions. They plot the max attention logit along with training steps. Practical observations and findings are very valuable and important. However, our work devotes to mathematically analyze why the spectral energy of $\boldsymbol{W}_q^{\top} \boldsymbol{W}_k$ will concentrate to a few directions and why the spectral energy concentration will lead to model crash.
> > >
> > > > Our paper, by a rigid theoretical analysis, reveals that the Spectral Energy Concentration (SEC) of $\boldsymbol{W}_q^{\top} \boldsymbol{W}_k$ makes the attention map matrix to be sparse yet low-rank and it is the inner reason leading to model crash. This is totally different from [1]. We believe that both practical and theoretical research are valuable and important. Our findings with rigid theoretical justification are of meaningful contributions and worthwhile to publish.
> > >
> > > > In addition, the mentioned "smooth rank measure" sounds very interesting and is appealing to be investivated as future work. Thanks the reviewer for the valuable suggestion.
> > >
> > > > [1] Wortsman, Mitchell, et al. "Small-scale proxies for large-scale transformer training instabilities." arXiv preprint
> > > arXiv:2309.14322 (2023).
> > >
> > > I agree with the authors that the paper takes an analytical approach and proposes a measure, SEC, to quantify an optimization failure. The problem that I observe with the proposed measure that a hyperparameter, s, needs to be specified for calculating SEC. Hence the suggestion to use a smooth rank measure that avoids having to search over a value but instead capture a notion of rank that could be ``good enough'' in practice. My viewpoint is that SEC has a small limitation that can be addressed with known tools. If the authors agree then suggestion is to clarify this point in the paper. If not, forgive my misunderstanding!
> > >
> > > A separate note: Since the authors tackle  optimization issues the following paper that tracks sharpness during optimization [Gimer] maybe of interest to the authors
> > > (Gilmer] [Gilmer et al.]([https://arxiv.org/abs/2110.04369)
> > >
> > > I will update my score to reflect my revised outlook on the paper

---

> > > > ### Author Response · Authors · 2024-11-26
> > > > **Thank you for your kind reply**
> > > >
> > > > Thank you for your positive feedbacks. Your insightful and constructive comments really help us improve the quality of our work. Thanks again for your suggestions about the discussion and the reference. We will consider them in the updated version of our paper.

---

### Official Review · Reviewer_QdMW · 2024-11-04

**Soundness:** 3
**Presentation:** 2
**Contribution:** 3
**Rating:** 8
**Confidence:** 3

**Summary:**

This paper deals with the challenges of scaling Transformers without relying on techniques like learning rate warmup. It provides a theoretical analysis that identifies the concentration of spectral energy in weight matrices as a key factor behind model crashes, leading to entropy collapse. To resolve this issue, the authors propose a novel optimization strategy that smooths weight updates, preventing rapid concentration of spectral energy. Extensive experiments with various Transformer models, including ViT, Swin Transformer, and GPT, demonstrate that this approach effectively trains models.

**Strengths:**

This paper demonstrates strong originality, with the necessary mathematical definitions and proofs included as required. It is well-structured and organized clearly. The essential mathematical framework is presented both in the main text and supplementary material. Additionally, experiments have been conducted to verify the theoretical claims made in the paper.

**Weaknesses:**

This paper has several areas for improvement in terms of writing, particularly regarding the use of mathematical notations in the abstract without proper definitions. A similar issue is evident in the Introduction as well.

The experiments are inadequate; the paper should incorporate additional baseline models for comparison, as it currently only discusses or modifies three models. It would be advantageous to present empirical results using well-known recent models, such as those with linear complexity in attention, like Linformer or Flatten Transformers.

 It would be preferable to provide proper references instead of linking to internet sources (such as Wikipedia) as seen on page 6 and elsewhere.

**Questions:**

I have a few questions and suggestions as follows.

1. In the abstract, there are mathematical notations that should either be predefined or the abstract should be written more clearly without them.

2. In the Introduction, some notations are defined later in the text, such as "$\delta vec(P)$" mentioned in the second contribution point, which should be defined earlier.

3. The paper claims that spectral energy concentration is a key issue behind model crashes. If attention is computed without using softmax (for example, in Linear attention models like Flatten or Vicinity Vision Transformer), will the same crash occur? Is the problem specifically due to softmax, which might cause sparsity or model instability?

4. The paper states that "the attention map is a sparse yet low-rank matrix" (in the third contribution point). Does this imply that "$A = softmax(\frac{P}{sqrt{d_q}})$" is sparse? If so, please provide a reference or proof for this assertion.

5. In the experimental results, please clarify the choice of using only pure encoder architectures (like ViT) or pure decoder architectures (like GPT). Is it possible to utilize an architecture that combines both encoder and decoder components?

6. It would be preferable to provide proper references instead of linking to internet sources (such as Wikipedia) as seen on page 6 and elsewhere.

7. Please include some recent variants of Transformers in the empirical results for comparison.

---

> ### Author Response · Authors · 2024-11-23
> **Author responses to Reviewer QdMW (part 1/3)**
>
> We would like to express our sincere thanks to the reviewer for appreciating the contributions of this paper as __"strong originality, with the necessary mathematical definitions and proofs included as required"__. While we observe more and more researchers developing fantastic applications, e.g., ChatGPT, Stable Diffusion, and Sora,  based on Transformer models, the mathematical understanding of the research community to these architectures is still insufficient. We are very eager to contribute to deepening the theoretical understanding of Transformer models. Your recognition of our work means a lot to us, and we will persist in this research direction.
>
> We have added more experimental results and revised the paper according to your suggestions. Hope our revisions and responses resolve your concerns.
>
> &nbsp;
>
> *__1.1 Reviewer's comments on notation and reference__*
>
> > *This paper has several areas for improvement in terms of writing, particularly regarding the use of mathematical notations in the abstract without proper definitions. A similar issue is evident in the Introduction as well.*
>
> __Our response__: we would like to express our sincere thanks to the reviewer for your professionalism and rigorousness. According to your suggestions, we have made the following revisions.
>
> 1. We added the definition for $\boldsymbol{W}_q$ and $\boldsymbol{W}_k$ in the abstract when we mention them;
>
> 2. We added the defintion for $\boldsymbol{P}$ when we mention it in the introduction;
>
> 3. We supplied the references for Vectorization and Kronecker product instead of refering them to wiki pages.
>
> &nbsp;
>
> *__1.2 Reviewer's comments on experiments__*
>
> > *The experiments are inadequate; the paper should incorporate additional baseline models for comparison, as it currently only discusses or modifies three models. It would be advantageous to present empirical results using well-known recent models, such as those with linear complexity in attention, like Linformer or Flatten Transformers.*
>
> __Our response__: thank you very much for your suggestion. In the revised paper, we have supplied the experiments using Flatten Transformers, and meanwhile, we also conducted experiments on larger GPT and ViT. Specifically, we conducted ViT-g, which has 1 billions parameters (116 times of the parameters in ViT-base). We also conducted experiments on GPT-large. These experiments have shown that our proposed solution can stablize the training processs without using learning rate warmup; whereas the original Flatten-Transnformer uses 20 epochs of warmup.
>
> We added the experiments on Flatten-Transnformer in **Appendix L** and the experiments on larger GPT and ViT in **Appendix J and K** in the revised verion of our submission.
>
> &nbsp;
>
> *__1.3 Reviewer's comments on references__*
>
> > *It would be preferable to provide proper references instead of linking to internet sources (such as Wikipedia) as seen on page 6 and elsewhere.*
>
> __Our response__: Sure, thank you very much for your suggestions. We have supplied proper references for the Kronecker product  and the Vectorization in the revised paper. The reference [1] provides the detailed introduction to Kronecker product and Vectorization.
>
> [1] Graham, Alexander. *Kronecker products and matrix calculus with applications*. Courier Dover Publications, 2018.
>
> &nbsp;
>
> ***2.1 Reviewer's questions 1,2,and 6 on notations and references***
>
> __Our response__: see above.
>
> &nbsp;
>
> ***2.3 Reviewer's question 3 on spetral energy concentration***
>
> 3. > *The paper claims that spectral energy concentration is a key issue behind model crashes. If attention is computed without using softmax (for example, in Linear attention models like Flatten or Vicinity Vision Transformer), will the same crash occur? Is the problem specifically due to softmax, which might cause sparsity or model instability?*
>
> **Our response**: Thank you very much for your comment. This is an insightful question. If we do not use softmax, the attention map is computed as $\boldsymbol{A} = \boldsymbol{P} = \boldsymbol{X}^{\top} \boldsymbol{W}_q^{\top} \boldsymbol{W}_k \boldsymbol{X}$.
> According to the gradient computation, we have the Jacobian matrix as follows:
>
> $$
> \frac{\partial \operatorname{vec}(\boldsymbol{P})}{\partial \operatorname{vec}({\boldsymbol{W}_q}^{\top}{\boldsymbol{W}_k} )} = \boldsymbol{X}^{\top} \otimes \boldsymbol{X}^{\top}
> $$
>
> If $\boldsymbol{X}$ is low-rank, it will still triger $\boldsymbol{W}_q^{\top} \boldsymbol{W}_k$ to be low-rank. Since we do not use softmax, $\boldsymbol{A}$ will not a sparse yet low-rank matrix. Instead, it will only be low-rank but not sparse.
>
> &nbsp;

---

> > ### Author Response · Authors · 2024-11-23
> > **Author responses to Reviewer QdMW (part 2/3)**
> >
> > ***2.4 Reviewer's question 4 on sparse yet low-rank matrix***
> >
> > 4. > *The paper states that "the attention map is a sparse yet low-rank matrix" (in the third contribution point). Does this imply that "" is sparse? If so, please provide a reference or proof for this assertion.*
> >
> > **Our response**: thank you very much for your comments. We are sorry for the confusion. To clarify this point, we would like to present a few examples below. Given three matrices, say, $\boldsymbol{A}, \boldsymbol{B}$ and $\boldsymbol{C}$, which are defined as follows:
> >
> > $\boldsymbol{A} = \begin{pmatrix} \frac{1}{5} & \frac{1}{5} & \frac{1}{5} & \frac{1}{5}& \frac{1}{5}\\\\ \frac{1}{5} & \frac{1}{5} & \frac{1}{5} & \frac{1}{5}& \frac{1}{5}\\\\ \frac{1}{5} & \frac{1}{5} & \frac{1}{5} & \frac{1}{5}& \frac{1}{5}\\\\ \frac{1}{5} & \frac{1}{5} & \frac{1}{5} & \frac{1}{5}& \frac{1}{5}\\\\ \frac{1}{5} & \frac{1}{5} & \frac{1}{5} & \frac{1}{5}& \frac{1}{5} \end{pmatrix}$,$\boldsymbol{B} = \begin{pmatrix} 1 & 0 & 0 & 0& 0\\\\ 0 & 1 & 0& 0& 0\\\\0 & 0 & 1& 0& 0\\\\ 0 & 0 & 0& 1& 0\\\\0 & 0 & 0& 0& 1  \end{pmatrix}$,  $\boldsymbol{C} = \begin{pmatrix} 1 & 0 & 0 & 0& 0\\\\ 1 & 0 & 0& 0& 0\\\\ 1 & 0 & 0& 0& 0\\\\ 1 & 0 & 0& 0& 0\\\\1 & 0 & 0& 0& 0 \end{pmatrix}$
> >
> > We can see that $\boldsymbol{A}$ is low-rank, $\boldsymbol{B}$ is sparse but not low-rank, $\boldsymbol{C}$ is sparse and simultaneously low-rank.
> >
> > In previous papers [1,2], researchers have analyzed the problem of model crash via *rank collapse of activations and entropy collapse of attention map*. Dong et al. [1] attributes the model crash into *rank collapse* of the activations, but in [2], Zhai et al. argue that it is an *entropy collapse* of the attention map which leads to the model crash.
> >
> > However, based on our analysis and a number of empirical observations, we found many counterexamples to the entropy collapse, and meanwhile the rank collapse of the activation cannot fully describe the inner reason of the model crash (which is rooted from the weight matrix instead of activations).  When the state of $\boldsymbol{C}$ usually happens, the model crash,
> >
> > - _Rank collapse_ of the activations cannot reveal the underlying cause that exists in the weight matrix.
> >
> > - $\boldsymbol{B}$ is a counterexample of _entropy collapse_. we observe that in some successful cases, state $\boldsymbol{B}$ occurs. According to the definition of entropy collapse, state $\boldsymbol{B}$ should lead to model crash; however, our experiments show that the model remains stable in this state.
> >
> > - Attention matrix being sparse yet low-rank is the state of the attention map when a model crashs. We believe that the rank collapse of activations and the entropy collapse of attention map are not precise enough to characterize the state of the model crash. According to our analysis, we argue that Spectral Energy Concentration (SEC) of $\boldsymbol{W}_q^{\top} \boldsymbol{W}_k$ is the inner reason behind the model crash, and it leads a sparse yet low-rank attention matrix which is exactly the phenomena we observed on the attention matrix.
> >
> > When we mention of a sparse yet low-rank matrix, we mean that the matrix is sparse and at meantime also low-rank. Such a concept can be found in a recent textbook [3]. Be breif, a sparse yet low-rank matrix is exceptional rare case for which the well-established theories for matrix completion and recovery must exclude (see Chapter 4 in [3]). However such a concept is also of practical value in computer vision (see Chapter 15 in [3]).
> >
> > [1] Yihe Dong, Jean-Baptiste Cordonnier, and Andreas Loukas. "Attention is not all you need: Pure attention loses rank doubly exponentially with depth." *International Conference on Machine Learning*. PMLR, 2021.
> >
> > [2] Shuangfei Zhai, et al. "Stabilizing transformer training by preventing attention entropy collapse." *International Conference on Machine Learning*. PMLR, 2023.
> >
> > [3] John Wright, and Yi Ma. *High-dimensional data analysis with low-dimensional models: Principles, computation, and applications*. Cambridge University Press, 2022.
> >
> > &nbsp;

---

> > > ### Author Response · Authors · 2024-11-23
> > > **Author responses to Reviewer QdMW (part 3/3)**
> > >
> > > ***2.5 Reviewer's question 5 on encoder-decoder architecture***
> > >
> > > 5. > *In the experimental results, please clarify the choice of using only pure encoder architectures (like ViT) or pure decoder architectures (like GPT). Is it possible to utilize an architecture that combines both encoder and decoder components?*
> > >
> > > **Our response**: Thank you for your comments. Theoretically, the proposed method can be applied to a encoder-decoder architecture, such as machine translation, object detection (DETR), and T5.  We will verify the proposed method on a hybrid architecture in the future. Due to time limitation, we cannot provide results in this revision during the rebuttal period, but we still appreciate your suggestion very much.
> > >
> > > &nbsp;
> > >
> > > ***2.7 Reviewer's question 7 on some other variants***
> > >
> > > 7. > *Please include some recent variants of Transformers in the empirical results for comparison.*
> > >
> > > **Our response**: In the revised paper, we have supplied experiments using Flatten Transformers, and meanwhile, we also conducted experiments on larger GPT and ViT. Experimental results further verify our understanding to the model crash of Transformer.
> > >
> > > We sincerely thank the reviewer QdMW for your valuable suggestions, which have greatly helped improve our paper. **We hope we have addressed your concerns above. We are open to have any further discussion.**
> > >
> > > &nbsp;

---

### Author Response · Authors · 2024-11-23
**Summary of our revisions**

We would like to thank all reviewers for their time and effort in reviewing our paper!

To address reviewers' comments and concerns, we have made the following changes:

- We added experiments on a larger vision Transformer, i.e., StableViT-giant which has 1B parameters, in **Appendix J**.

- We added experiments on a larger language Transformer, i.e., StableGPT-large which has 774M (0.774B) parameters, in **Appendix K**.

- We added experiments on a Flatten Transformer in **Appendix L**.

- We visualized the actual step size for different weights in different layers over time in **Appendix M**.

- We revised the paper, updated the figures, clarified some concepts when we first mentioned them according to the reviewers' suggestions.

The revised contents are highlighted in purple in the revised paper.  **We kindly invite all reviewers to take a look at our new experimental results!**

We sincerely thank all reviewers again for their valuable suggestions, which have greatly helped strengthen our paper.

If you have any further questions, we would be happy to discuss them!

---

### Public Comment · ~Atli_Kosson1 · 2024-11-24
**Related Work on Warmup**

Dear Authors,

I read your ICLR submission with great interest. I think it is a good contribution that provides an interesting perspective on warmup. I would like to draw your attention to two of our recent works which I think may be quite relevant. In our [NeurIPS 2024 paper](https://arxiv.org/abs/2410.23922) (which is parallel work with yours), we explore the need for warmup in GPT training. One of our findings is that the need for warmup can be significantly reduced and sometimes fully eliminated by limiting the Frobenius norm of the update relative to that of the weights. This differs from your method which is based on the spectral norm instead of the Frobenius norm. Another difference is that we control the size of the update throughout training instead of clipping it, which can be done without any additional hyperparameters (for Adam / Lion like optimizers). The first paper expanded upon a similar observation we made in our earlier [ICML 2024 work](https://arxiv.org/abs/2305.17212) on how weight decay affects the training dynamics of neural networks. Would you perhaps consider discussing these as related work in a future revision?

Thank you,

Atli Kosson

---

> ### Author Response · Authors · 2024-11-25
> **Thank you for your interests and recognizing of our work.**
>
> Thank you for your interests and recognizing of our work. We will read your recommended works and try to add some discussion on these works.

---

### Meta-Review · Area_Chair_WTqT · 2024-12-16

**Metareview:**

This paper aims to explain the mechanism behind warump in Transformer training.
It starts with a visualization of a two-layer Transformer in Section 3 by monitoring several quantities, and then develops a metric spectral energy concentration based on the sigular value of $W_q^{T} W_k$. Hence a smooth weight updating rule is proposed for warmup. The extensive experiments support the effectiveness.

The AC suggests the authors to conduct more experiments on relatively deep Transformers (e.g., three layers) to verify the experimental findings. Besides, the proposed spectral engergy concentration is quite close to effective rank used in machine learning theory, e.g., benign overfitting. This also deserves to explore. Frankly speaking, massive known results are delivered in the main text, e.g., Proposition 1, Theorem 2 (Weyl inequality), it would be best to put them into the appendix.

**Additional Comments On Reviewer Discussion:**

All of the reviewers engage with the discussion and the authors address most of concerns, e.g., adding more experiments. I expect that these results will be included into the paper.

---

### Decision · Program_Chairs · 2025-01-22

Accept (Poster)